# Common and rare genetic variants show network convergence for a majority of human traits

Sarah N Wright [iD][1], Jane Yang[1] & Trey Ideker [iD][1,2]✉

## Abstract

**While both common and rare variants contribute to the genetic etiology of complex traits, whether their impacts manifest through the same effector genes and molecular mechanisms is not well understood. Here, we systematically analyze common and rare variants associated with each of 373 phenotypic traits within a large biological knowledge network of gene and protein interactions. While common and rare variants implicate few shared genes, they converge on shared molecular networks for more than 75% of traits. We demonstrate that the strength of this convergence is influenced by core factors such as trait heritability, gene selective constraint, and tissue specificity. Using neuropsychiatric traits as examples, we show that common and rare variants impact genes with shared functions across multiple levels of biological organization. These findings underscore the importance of integrating variants across the frequency spectrum and establish a foundation for network-based investigations of the genetics of diverse human diseases and phenotypes.**

**Keywords** Complex Traits; Network Biology; Common Variants; Rare Variants; Genetic Architecture
**Subject Category** Genetics, Gene Therapy & Genetic Disease

## Introduction

Genome-wide association studies (GWAS) have identified thousands of common variants (CVs) that are significantly associated with complex human traits, including a wide array of diseases and clinical biomarkers (Abdellaoui et al, 2023). However, elucidating causal mechanisms impacted by these CVs has proven challenging because they typically fall in non-coding regions and exert small effect sizes. Therefore, GWAS have been recently complemented by large-scale exome and whole-genome sequencing efforts, which have enabled the identification of trait-associated variants present at low frequencies in coding regions (Karczewski et al, 2022). The resulting rare variants (RVs) exert larger effect sizes (Fiziev et al,

2023) and directly implicate genes and biological pathways (Momozawa and Mizukami, 2021). Separately, these common and rare variant association studies have identified important genetic factors underlying human diseases and other complex traits (Abdellaoui et al, 2023); however, their joint influence remains poorly understood.

There is growing evidence that RVs contribute to trait heritability and that incorporating them alongside CVs can improve polygenic risk score (PRS) predictions (Wainschtein et al, 2022; Fiziev et al, 2023). However, the contribution of RVs appears to be highly trait-dependent. For example, Wainschtein et al found that RVs in low linkage disequilibrium regions significantly drive variance in human height (Wainschtein et al, 2022), and Fiziev et al showed that rare-variant PRS are better at identifying individuals at phenotypic extremes across 78 phenotypes (Fiziev et al, 2023). Conversely, minimal contributions from RVs have been reported for the polygenic risk prediction of metabolic traits (Kim et al, 2022b). Moreover, global heritability and polygenic risk score analyses do not necessarily provide insights into the biological mechanisms impacted by genetic variation.

Previous studies have shown that CV and RV associations can systematically prioritize different genetic loci and effector genes (Spence et al, 2026). These differences have been linked to challenges such as limited statistical power, differing selection pressures, population stratification, or heterogeneity in clinical manifestations (Zhou et al, 2023a; Wray et al, 2018; Gettler et al, 2021). CVs and RVs at different genetic loci, however, do not necessarily imply functional divergence. For example, for schizophrenia (Chang et al, 2018) and autism spectrum disorder (Ben-David and Shifman, 2012), such variants have been shown to converge onto relevant molecular networks that implicate protein complexes and biological pathways. However, to date, such systems-level concordance has been demonstrated in only a small number of human phenotypes.

Here, we examine the role of common and rare variants across the human phenome via a systematic network integration of CVs and RVs for 373 distinct traits. We identify significant network convergence for 283 phenotypes, 84 of which have no shared common and rare variant-associated genes. In so doing, we aim to understand the factors that drive differences in the discovered CVs and RVs and their overall network convergence.

---

[1]Department of Medicine, University of California, San Diego, La Jolla 92093 CA, USA. [2]Institute for Genomic Medicine, University of California, San Diego, La Jolla 92093 CA, USA. ✉E-mail: tideker@health.ucsd.edu

# Results

## Identification of 373 human traits with common and rare variant associations

We identified phenotypes and associated common and rare variants from studies curated by the GWAS Catalog (Sollis et al, 2023; Data ref: GWAS Catalog Associations, 2025) and the Rare Variant Association Repository (RAVAR) (Cao et al, 2024; Data ref: RAVAR Associations, 2024) (Fig. 1A). Common-variant-associated genes (CVGs) were defined as distinct genes with a trait-associated single-nucleotide polymorphism (SNP) located within the gene region defined by Ensembl (Sollis et al, 2023; Harrison et al, 2024). After quality control (Methods), we identified 7538 studies with at least three suggestive CVGs ($p < 1 \times 10^{-5}$), representing 2339 distinct traits. Separately, we sourced rare-variant-associated genes (RVGs) from gene-based rare variant studies in the RAVAR database. We identified 2207 studies representing 1121 distinct traits with at least three suggestive RVGs ($p < 1 \times 10^{-4}$). From these studies, we identified 373 traits with both common and rare variant association results.

This collection of traits represented a variety of clinical biomarkers, disease states, and other complex phenotypes (Fig. 1B). Among categorical traits ($n = 187$), neoplasms and cardiovascular diseases were the most represented biological domains, while lipid and other metabolic measurements were the most represented domains among continuous traits ($n = 186$). Study populations varied from large biobank cohorts, such as the UK Biobank, to smaller ascertained studies. Rare variant study populations tended to be larger, with a median of 210,000 individuals, compared to a median of 136,000 individuals for common variant studies (Fig. 1C, q = $5.8 \times 10^{-8}$). CVGs were more mutationally constrained (q = $3.5 \times 10^{-29}$), as evidenced by higher values of $s_{het}$, a continuous metric that measures the reduction in fitness for heterozygous carriers of LOF mutations in a given gene (Zeng et al, 2024; Data ref: GeneBayes, 2023). In addition, CVGs had longer gene regions (q = $1.6 \times 10^{-50}$) and more functional annotations (q = $2.2 \times 10^{-5}$), with expression in a greater number of tissues (q = $4.9 \times 10^{-4}$). The differences between CVGs and RVGs were consistent when examining alternative metrics of gene length and mutational constraint (Fig. EV1A). For example, CVGs had longer CDS sequences (q = $8.4 \times 10^{-9}$) and showed stronger evolutionary conservation (phyloP, q = $4.2 \times 10^{-11}$). These distinct gene properties likely reflect a combination of differences in variant impacts, as well as selection biases from the sequencing and statistical association methodologies.

## Common and rare variant associations implicate few shared genes

Identifying a gene via both CV and RV analyses can lend confidence to a gene-trait association. Overall, continuous traits identified a greater number of shared CV/RV-associated genes than categorical traits (Fig. 1D). The greatest agreement was observed for HDL cholesterol change measurement, with 11 shared genes among 22 CVGs and 17 RVGs (q = $2.2 \times 10^{-28}$). Comparing the association p-values of shared genes to all common-only and rare-only associations, we found that the shared gene associations tended to be more significant than the disjoint associations (Fig. 1E), supporting their role as high-confidence trait associations. While 179 traits had significantly more shared genes than expected by chance (Fig. 1F), the shared genes represented a minority of all CVGs and RVGs for these traits (median of 28% and 18%, respectively). Furthermore, of the 373 phenotypes, 189 did not generate any shared genes. The minimal correspondence between CVGs and RVGs for a large proportion of traits suggested that focusing solely on shared genes does not provide a complete picture of trait genetics.

## A majority of traits demonstrate network convergence of common and rare variant associations

To move beyond genes and investigate shared functions, we integrated CVGs and RVGs with a comprehensive biological knowledge network, the Parsimonious Composite Network (PCNet 2.0) (Wright et al, 2025). This network contains 3.85 million pairwise relationships among human genes and proteins derived from a variety of evidence sources such as protein–protein interactions, co-expression, and known biological pathways. Using an approach called Network Colocalization (Rosenthal et al, 2023, 2021) (NetColoc, Fig. 2A), we propagated the CVG and RVG association scores throughout the network. From each trait's propagated scores, we formed trait-specific networks that captured the subset of genes in close network proximity to both CVGs and RVGs. To quantify overall network convergence, we defined the COLOC score as the observed over expected size of each trait-specific network.

Of the 373 traits, 254 showed significant network convergence of CVGs and RVGs (Fig. 2B), suggesting a high concordance in the molecular systems impacted by common and rare variants present in gene regions. Of these, 174 traits produced trait-specific networks at least two times larger than expected by chance (COLOC score ≥ 2). Continuous traits showed significantly stronger CVG/RVG convergence ($p = 1.0 \times 10^{-54}$, Fig. 2C), consistent with the higher number of shared genes observed for these traits. Nonetheless, 74 categorical traits had significant convergence, with a diverse set of conditions represented in the top 10, including myeloproliferative disorder, Crohn's disease, and allergic rhinitis.

Network convergence was particularly strong among lipid measurement traits, which had trait-specific networks that were, on average, 3.6 times larger than expected by chance. In contrast, infection, neoplasm, and skeletal traits exhibited low rates of network convergence, with significant COLOC scores for only 4 of 18, 9 of 26, and 6 of 17 traits, respectively (Fig. 2D). The biological domains with stronger convergence contained a higher proportion of continuous traits, and these traits tended to have a higher number of shared genes ($r_s = 0.82$, Fig. EV1B). The identification of some shared genes could be driven by synthetic associations (Dickson et al, 2010), whereby a CV tags nearby causal RVs, rather than by the existence of independent causal RVs and CVs. When treating all shared genes as synthetic associations for a subset of 11 traits, we found that significant convergence was maintained for 9 of the tested traits, but with decreased COLOC scores (Fig. EV1C). This decrease in colocalization strength is expected, as shared CVGs and RVGs (whether true independent signals or synthetic

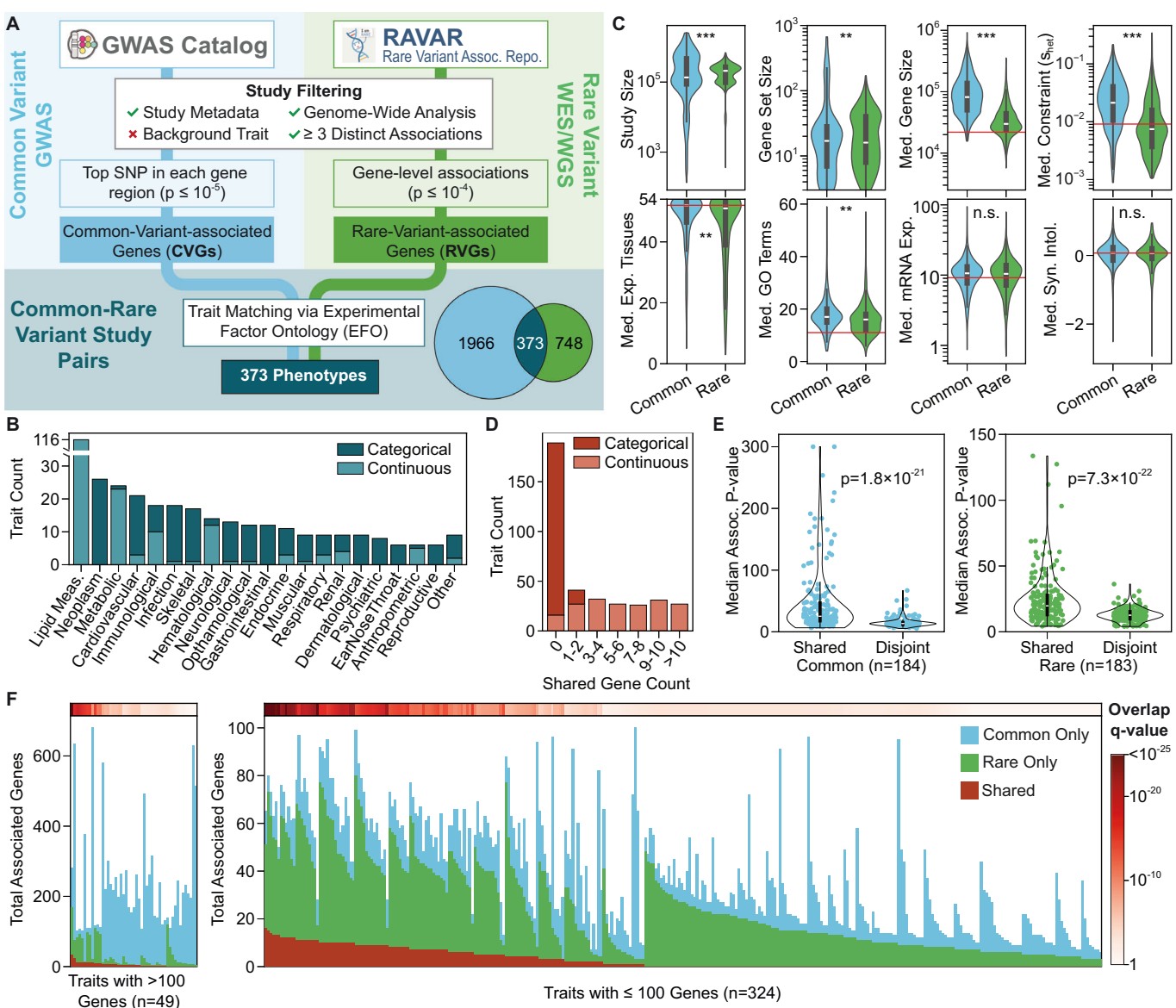

**Figure 1. Curation and analysis of common and rare variant associations for 373 traits.**

(A) Overview of the data curation and trait-matching pipeline for defining sets of common-variant-associated genes (CVGs) and rare-variant-associated genes (RVGs). (B) Distribution of traits across biological domains and trait classifications. Domains with fewer than five traits are included in 'Other'. (C) Comparison of CVG and RVG properties across 373 traits (Wilcoxon Signed-Rank test, BH correction). For gene-level properties, the value for each set of CVGs or RVGs was taken as the median of all genes in each set. Red lines indicate the median value for all protein-coding genes, where applicable. ***q < 1 × 10⁻⁵, **q < 1 × 10⁻³, n.s. q > 0.05. $q_{StudySize} = 5.8 \times 10^{-8}$, $q_{GeneSetSize} = 7.5 \times 10^{-4}$, $q_{MedGeneSize} = 1.6 \times 10^{-50}$, $q_{MedConstraint} = 3.5 \times 10^{-29}$, $q_{MedExpTissues} = 4.9 \times 10^{-4}$, $q_{MedGoTerms} = 2.2 \times 10^{-5}$, $q_{MedmRNAExp} = 0.47$, $q_{MedSynIntol} = 0.56$. (D) Distribution of the number of shared genes identified per trait as both CVGs and RVGs, stacked by trait classification. (E) Comparison of the median CVG (left) and RVG (right) association p-values between shared genes and disjoint genes (Wilcoxon Signed-Rank test). (F) Stacked bar chart of the total number of distinct genes associated with each trait. Shared genes are those identified as both CVGs and RVGs. The top heatmap shows the q-value of the number of shared genes (hypergeometric test, BH correction). Plot is split by traits with >100 associated genes and traits with ≤100 associated genes for visualization. Data information: All violin plots extend to the minimum and maximum observations, with center box plots showing the median property value and interquartile range (IQR), with the lower and upper whiskers extending to Q1 − 1.5IQR and Q3 + 1.5IQR. Source data are available online for this figure.

associations) strongly reinforce each other within the network analysis. Given the low expected frequency of synthetic associations (Anderson et al, 2011; Wray et al, 2011; Orozco et al, 2010), treating all shared genes as synthetic associations provides a conservative upper bound on their potential impact on CVG/RVG convergence.

## Network convergence is robust across variable study designs

To test generalizability across studies, we performed additional NetColoc analyses using 223 of the 373 traits for which multiple studies of common or rare variants were available. Up to four

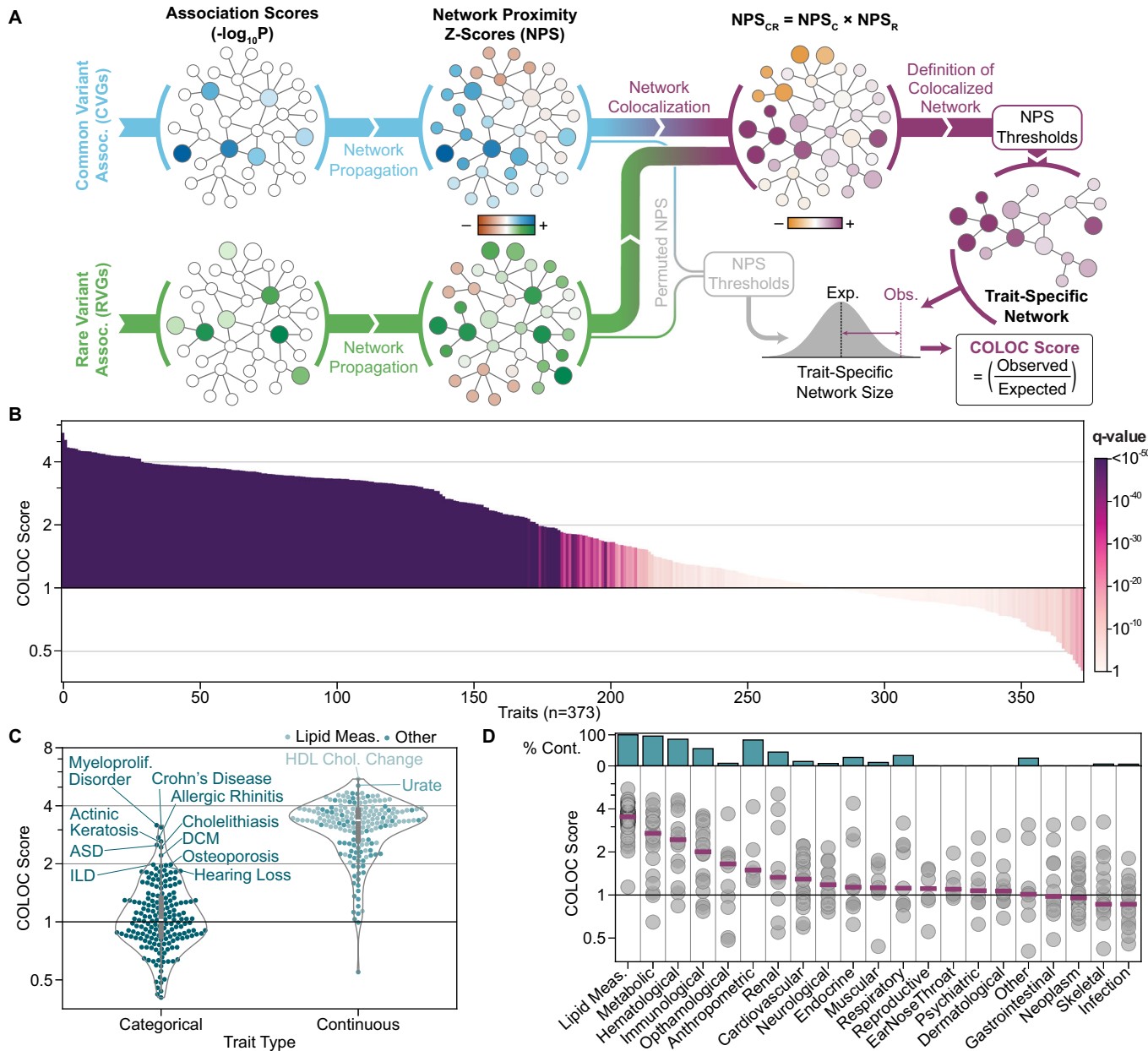

**Figure 2. Network convergence of common and rare variant associations across 373 human traits.**

(**A**) Schematic of the Network Colocalization (NetColoc) pipeline for assessing the convergence of variant-associated genes and defining trait-specific networks. (**B**) Network convergence of common and rare variants for 373 traits. Traits are ordered by COLOC score, and bar color indicates the q-value of the COLOC score calculated from a Z-test comparing the observed trait-specific network size to 1000 permutations of the expected size, with BH correction. (**C**) Comparison of COLOC scores between categorical (n = 187) and continuous (n = 186) traits. The center box plots show the median property value and interquartile range (IQR), with the lower and upper whiskers extending to Q1 − 1.5IQR and Q3 + 1.5IQR. The violins extend to the minimum and maximum observations. ILD: interstitial lung disease, ASD: autism spectrum disorder, DCM: dilated cardiomyopathy. (**D**) Distribution of COLOC scores across biological domains. Purple lines show the median COLOC score per domain. The upper bar plot shows the percentage of domain traits that are continuous. The number of traits within each domain is displayed in Fig. 1B. Source data are available online for this figure.

additional CV studies and four additional RV studies were selected per trait (Fig. 3A), prioritizing those conducted in different population cohorts. COLOC scores for each trait were largely consistent across the various studies ($r_s$ = 0.84, Fig. 3B). Nonetheless, the variation amongst scores highlighted that the strength of CVG/RVG convergence observed can be influenced by study selection.

Some traits, such as Crohn's disease and cystatin C measurement, showed large discrepancies between different CV studies, while others, such as schizophrenia and eosinophil count, showed discrepancies between different RV studies (Figs. 3C and EV2A).

We assigned the final COLOC score as the maximum score observed for each trait (Fig. EV2B). Despite having fewer total

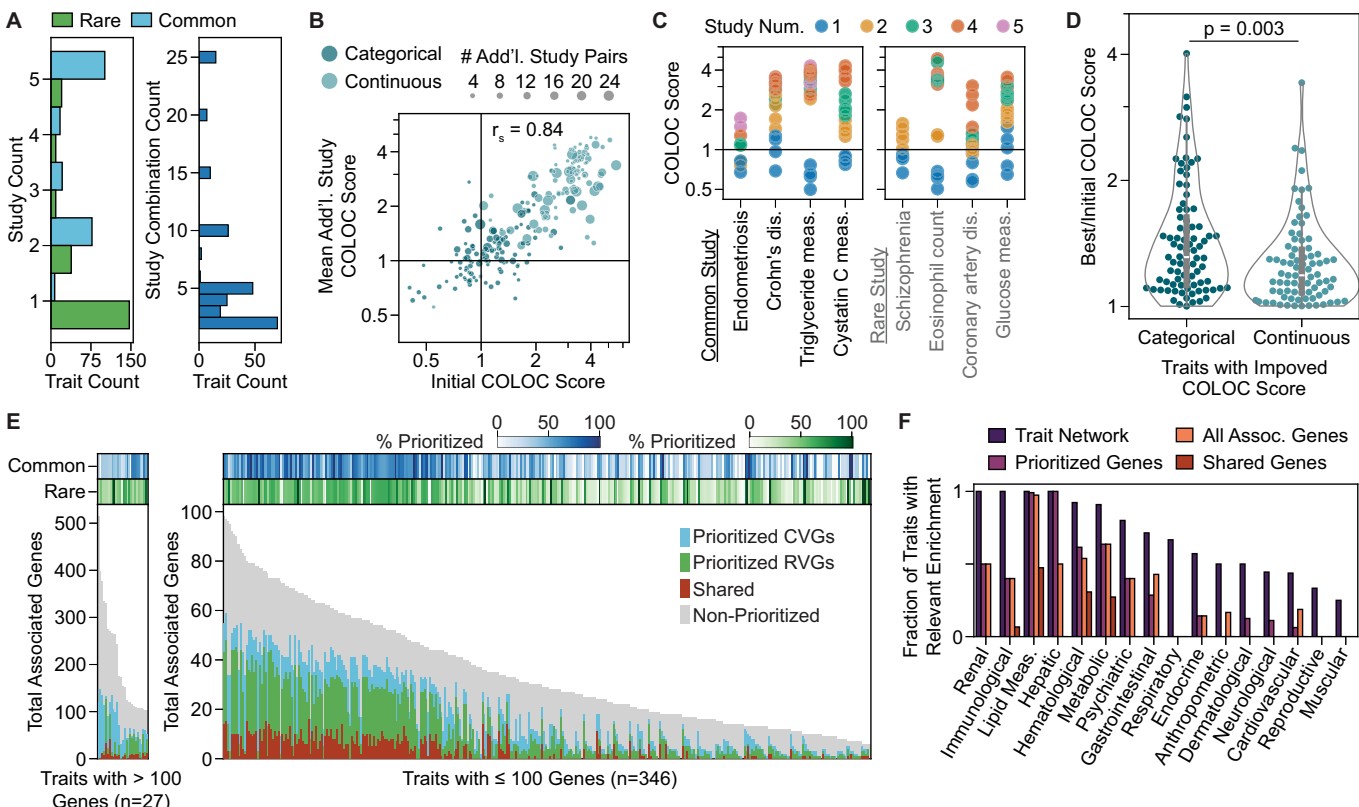

**Figure 3. Robustness of network colocalization and prioritization of trait associations.**

(A) Number of independent CV and RV studies tested per trait up to a maximum of five (left), and total number of common-rare study combinations analyzed per trait (right). (B) Initial COLOC scores (Fig. 2C), compared to the mean COLOC scores of all additional study combinations per trait. Results in (A) and (B) are shown for the 223 traits with at least one additional study combination. (C) COLOC scores for eight example traits with differential dependence on CV study selection (left, colored by CV study) and RV study selection (right, colored by RV study). (D) Ratio of best to initial COLOC scores across all common-rare study combinations analyzed per trait, compared across categorical ($n = 85$) and continuous traits ($n = 83$) using a Mann-Whitney U test. The best study pair per trait was selected as the one achieving the highest COLOC score. Traits with an improved COLOC score (best > initial) are shown. The center box plots show the median property value and interquartile range (IQR), with the lower and upper whiskers extending to Q1 − 1.5IQR and Q3 + 1.5IQR. (E) Prioritization of gene associations following network colocalization (stacked bar plot). Prioritized genes were defined as disjoint CVGs or RVGs included in the best-scoring trait-specific network for each trait. Top heat maps show the fraction of all CVGs and RVGs prioritized by the network analysis. (F) Fraction of traits per domain with at least one significant enrichment for genes expressed in a relevant tissue based on all genes in the trait-specific network (dark purple), prioritized input genes (purple), union of all CVGs and RVGs (orange), or only the shared CVGs and RVGs (red). Relevant tissues were defined manually for a subset of traits based on biological domains (Methods). The number of traits within each domain is displayed in Fig. 1B. See also Fig. EV2 for further analysis. Source data are available online for this figure.

studies, the improvement in score was higher for categorical traits ($p = 0.003$, Fig. 3D). From these results, we identified significant network colocalization for an additional 29 traits, bringing the total to 283 traits with convergent common and rare variant associations. Of these, 84 had no shared genes, reinforcing the power of a network approach for integrating non-overlapping, but complementary, genetic associations. Among the disjoint CVGs and RVGs, we defined prioritized genes as those included in the best-scoring trait-specific network for each trait. Overall, 44% of disjoint CVGs and 42% of disjoint RVGs were prioritized based on their global network proximity to other trait-associated genes (Fig. 3E). Compared to the original input genes, these prioritized trait genes and the full trait-specific networks showed better enrichment for genes expressed in trait-relevant tissues, demonstrating that the network approach prioritizes functionally relevant genes (Fig. 3F).

The observed convergence of CVGs and RVGs remained consistent when the network colocalization analysis was performed

using subcomponents of PCNet 2.0 (Fig. EV2C). The networks HumanNet v3 (Kim et al, 2022a), STRING (Szklarczyk et al, 2023), and the co-citation-free PCNet 2.2 (Wright et al, 2025) generated COLOC scores with Spearman correlations of 0.93, 0.92, and 0.95 to PCNet 2.0, respectively. Using an independent large network, OmniPath (Türei et al, 2021), we observed a slightly reduced performance with 68% of traits showing significant network convergence, compared to an average of 75% for PCNet 2.0 and its subcomponents (Fig. EV2D). Overall, while the global trend of CVG/RVG convergence is robust to network selection, the results for individual traits may vary based on the underlying network.

## Heritability and gene properties drive network convergence

Heritability and population prevalence are known to impact the discovery of common and rare variants. To examine the impact of

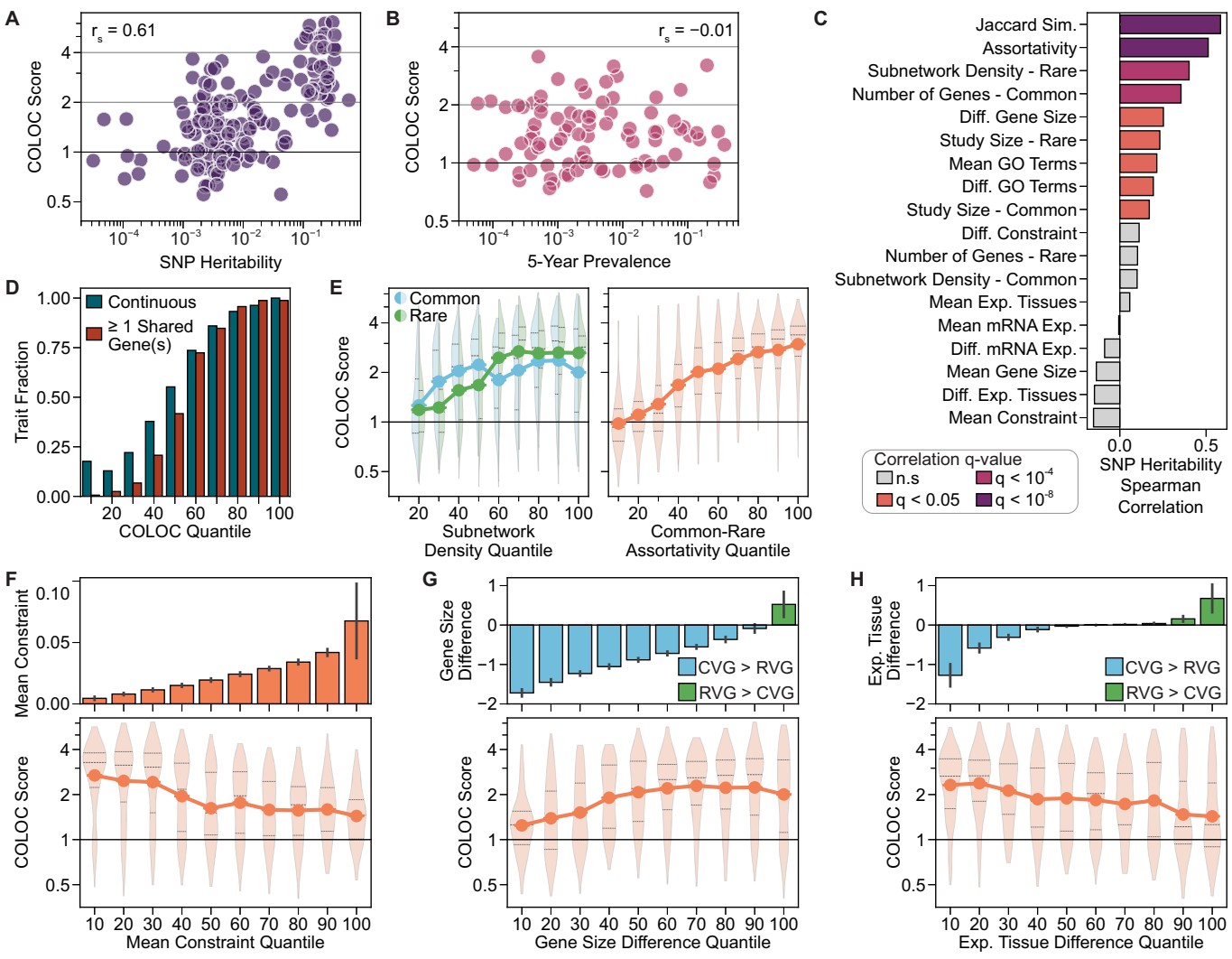

**Figure 4. Influence of trait and gene features on the convergence of common and rare variant associations.**

(A, B) Best COLOC score as a function of (A) trait SNP heritability ($n = 197$) and (B) trait prevalence ($n = 87$). Spearman correlation reported. Trait prevalence estimates represent averages over the five-year period from 2017 to 2021. (C) Spearman correlation of trait SNP heritability with 18 trait and gene features for the best study combinations for 197 traits with SNP heritability estimates. Jaccard similarity was calculated between CVGs and RVGs for each trait. (D) Fraction of traits within each COLOC score quantile that were continuous traits or had at least one shared CVG and RVG. (E–H) COLOC score as a function of (E) network properties of CVGs and RVGs, (F) mean selective constraint, (G) gene size difference, and (H) expressed tissue difference for CVGs and RVGs. In (D–H), values are calculated for all studies ($n = 1634$) of the 373 traits, including all combinations of primary and additional studies (Fig. 3). Point plots show the mean COLOC score for each feature quantile, and violin plots show the distribution of COLOC scores within each feature quantile, with the median and quartiles represented by horizontal lines. Violins extend to the minimum and maximum observed values. The first four quantiles represent $n = 164$ traits, with the remaining quantiles representing $n = 163$ traits, except for the leftmost plot of (E), where the quantiles represent $n = [330, 160, 165, 162, 172, 166, 152, 167, 160]$ traits due to repeated values. In (F–H), the upper bar plots show the mean value within each feature quantile. Error bars indicate the standard error. See also Figs. EV3 and EV4 for further exploration of heritability and gene set features. Data information: Gene features were summarized by the overall mean of CVGs and RVGs per trait, and the difference in means ($\delta$) between CVGs and RVGs ($\delta = 2(R - C)/(R + C)$). See also Fig. EV4B. Constraint: gene selective constraint defined by $s_{het}$ estimates. Exp. Tissues: Number of tissues with mRNA expression >1 TPM. Source data are available online for this figure.

these factors on the CVG/RVG network convergence, we collected SNP heritability estimates for 197 traits from the UK Biobank (Data ref: UKBB, 2022) and global prevalence estimates for 87 traits from the Global Burden of Disease study (Global Burden of Disease Collaborative Network, 2024; Data ref: GBD, 2021). The COLOC scores for these subsets of traits were strongly correlated with trait heritability ($r_s = 0.61$, Fig. 4A), but not population prevalence ($r_s = -0.01$, Fig. 4B). We further observed that classes of traits with the highest heritability, such as lipid

measurement and metabolic traits, also showed the strongest convergence of common and rare variant associations ($r_s = 0.83$, Fig. EV3A). To a lesser extent, the COLOC score was correlated with the number of associated genes, both within biological domains ($r_{CVG} = 0.51$, $r_{RVG} = 0.47$, Fig. EV3A) and globally ($r_{CVG} = 0.32$, $r_{RVG} = 0.28$, Fig. EV3B). These results suggest that both higher heritability and study power may increase the observed network convergence between common and rare variant associations.

**Table 1. Linear regression estimates of the influence of trait and gene features on the network convergence.**

| Feature | | Mean coefficient | 95% CI | q-value |
|---|---|---|---|---|
| Number of Genes | *Common* | 0.005 | (0.000, 0.016) | 0.38 |
| | *Rare* | −0.003 | (−0.011, 0.001) | 1.00 |
| Subnetwork Density | *Common* | **0.012** | (0.003, 0.021) | **0.02** |
| | *Rare* | **0.037** | (0.027, 0.047) | **0.02** |
| Study Size | *Common* | 0.001 | (−0.004, 0.007) | 1.00 |
| | *Rare* | −0.005 | (−0.012, 0.000) | 0.30 |
| Gene Constraint | *Mean* | **−0.013** | (−0.023, −0.002) | **0.02** |
| | *Difference* | 0.003 | (−0.002, 0.010) | 1.00 |
| Gene Size | *Mean* | 0.002 | (−0.004, 0.011) | 1.00 |
| | *Difference* | **0.014** | (0.005, 0.022) | **0.04** |
| Exp. Tissues | *Mean* | −0.001 | (−0.008, 0.005) | 1.00 |
| | *Difference* | **−0.011** | (−0.019, −0.001) | **0.02** |
| mRNA Expression | *Mean* | −0.002 | (−0.009, 0.003) | 1.00 |
| | *Difference* | 0.002 | (−0.003, 0.011) | 1.00 |
| GO Annotations | *Mean* | −0.005 | (−0.013, 0.000) | 0.72 |
| | *Difference* | 0.003 | (−0.001, 0.011) | 1.00 |
| No Shared Genes | | **−0.050** | (−0.066, −0.034) | **0.02** |
| Jaccard Similarity | | **0.122** | (0.109, 0.137) | **0.02** |
| Assortativity | | **0.039** | (0.028, 0.050) | **0.02** |
| Categorical Trait | | **−0.036** | (−0.048, −0.025) | **0.02** |

An Elastic Net regression model for predicting COLOC score was optimized via five-fold cross-validation, followed by bootstrap estimation of coefficients using 2000 bootstrap replicates over 1634 samples. Mean coefficient value, 95% confidence interval, and estimated q-value (Bonferroni corrected empirical *p*-value) are reported. Coefficients and q-values are shown in bold text for significant features (q < 0.05). Gene Constraint: gene selective constraint defined by $s_{het}$ estimates. Exp. Tissues: Number of tissues with mRNA expression >1 TPM.

Next, we investigated the influence of study design and gene properties on the extent of network convergence. For all traits, we collected 10 summary features covering the study type and population size, size and network properties of the CVG and RVG sets, and 10 summary values of biological properties, including gene expression, mutational constraint, and functional annotations (Fig. EV4A,B). The trait heritability showed widespread correlations with these features (Fig. 4C), offering potential explanations for how trait heritability influences the convergence of CVGs and RVGs. For example, traits with higher heritability had a larger number of CVGs and more interconnected RVGs. Using a linear regression model, we found that the trait and gene features explained 76% of the observed variance in COLOC score across all traits, driven by nine significant features (Table 1).

After adjusting for these features, the correlation between trait heritability and COLOC score was reduced (Fig. EV4C, $r_s = 0.16$), confirming that the study and gene properties can partly explain the heritability trend. As expected, the presence of shared genes, the trait type, and the interconnectedness of the CVGs and RVGs within the network were significant predictors of the network convergence results (Fig. 4D,E, Table 1). We further examined the relationship between the interconnectedness and COLOC scores across progressively randomized network structures, where interconnectedness was defined as the density of CVG and RVG subnetworks. As interactions in PCNet 2.0 were randomized, the subnetwork densities and the correlations between the subnetwork densities and the COLOC scores were reduced (Fig. EV4D). Therefore, the network structure of the CVGs and RVGs plays an important role in determining the overall convergence.

Independent of these factors, enhanced convergence was driven by sets of CVGs and RVGs containing more mutationally constrained genes and traits with smaller differences in gene size between CVGs and RVGs (Fig. 4F,G, Table 1). Interestingly, we found that the distribution of gene expression across tissues significantly influenced the observed COLOC score, as measured by the number of tissues with mRNA expression greater than 1 TPM. Traits with RVGs expressed in fewer tissues compared to CVGs showed significantly stronger network convergence (Fig. 4H, Table 1). Other trait and gene set features, such as the number of gene associations, the study population size, and the number of functional annotations, were not independently identified as significant predictors of CVG/RVG network convergence. However, due to widespread correlations between the features (Fig. EV4A), the reported coefficients and confidence intervals may partly reflect shared variance among related features. Therefore, the importance of properties such as the number of associations and the study population size cannot be ruled out.

## Tissue-specific RVGs are associated with stronger network convergence

Our results indicated that the network convergence of CVGs and RVGs was stronger for traits with RVGs expressed in a limited number of tissues compared to CVGs (Fig. 4H). To investigate the relevance of these genes, we performed a tissue-specific enrichment analysis for trait-relevant tissues, focusing on the traits exhibiting the largest differences in tissue distribution between the two sets of genes (Fig. 5A). Overall, traits with a significant enrichment for genes expressed in a relevant tissue had higher COLOC scores than traits without such an enrichment ($p = 0.004$, Fig. 5B). Furthermore, the fraction of gene sets with a relevant enrichment was greater for RVGs, compared to CVGs, indicating that RVGs with limited expression tended to be more consistent with the associated trait.

We then looked at results for glucose measurement (COLOC score = 3.6, Fig. 5C), the trait with the greatest tissue expression difference between CVGs and RVGs. Individually, CVGs associated with glucose measurement were expressed in a median of 53 tissues, compared to a median of 2 tissues for RVGs (Fig. 5A). Driven by three genes—G6PC2, SLC30A8, and KCNK16—the set of RVGs was exclusively enriched for pancreas-specific expression ($p_{adj} = 4.0 \times 10^{-5}$). The remaining two RVGs, ANKH and SFT2D3, were not specifically expressed in relevant tissues and were not prioritized as important by the network colocalization analysis. Overall, the ten glucose measurement-associated genes included in the glucose measurement network showed enrichment for relevant pathways such as glucagon signaling ($p_{adj} = 3.4 \times 10^{-5}$) and galactose metabolism ($p_{adj} = 4.7 \times 10^{-5}$), as well as human phenotypes such as increased waist-to-hip ratio ($p_{adj} = 1.0 \times 10^{-5}$) (Appendix Fig. S1). In contrast, the six non-prioritized trait-associated genes showed no significant enrichments for pathways or human phenotypes. However, it should be noted that the size of these gene sets may limit statistical power.

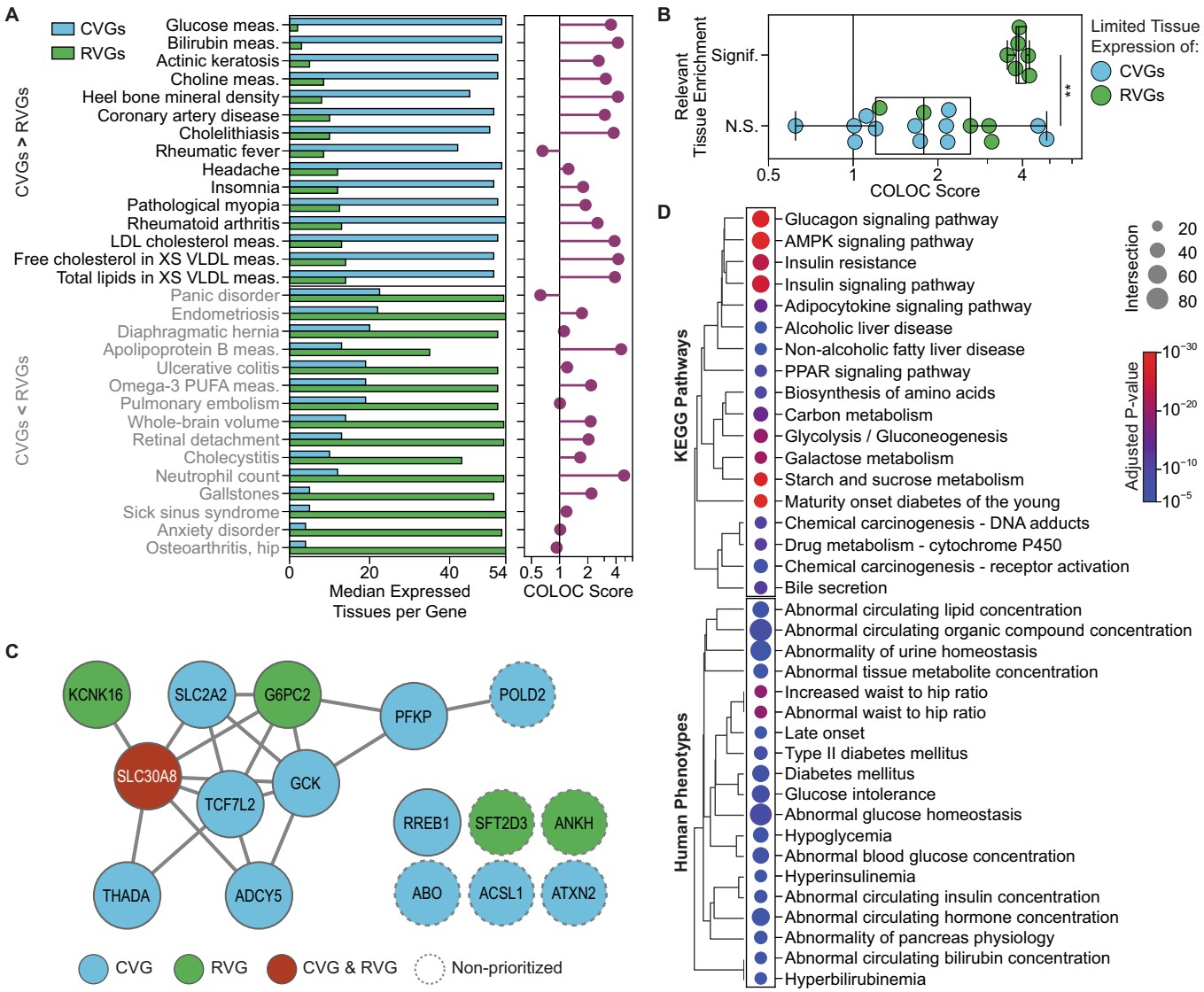

**Figure 5. Impacts of the tissue distribution of gene expression on the network convergence of common and rare variant associations.**

(A) Distribution of tissue expression for CVGs and RVGs, compared to the best COLOC score for 30 traits. Bar plot (left) shows the median number of tissues with HPA mRNA expression >1 TPM for CVGs and RVGs, and lollipop plot (right) shows the corresponding best COLOC score for each trait. Traits were selected by the difference between the mean number of expressed tissues for CVGs and RVGs ($\delta_{nTissues}$), to give the 15 traits with lowest $\delta_{nTissues}$ (RVGs expressed in fewer tissues) and the 15 traits with highest $\delta_{nTissues}$ (CVGs expressed in fewer tissues). (B) Comparison of the COLOC score to the tissue-specific enrichment in the most relevant tissue (Mann-Whitney U test, **$p = 0.004$). Significant tissue-specific enrichment was defined as an adjusted $p$-value < 0.05 (Methods) for the gene set (CVGs or RVGs) with the lower mean number of expressed tissues. Box plots show the median COLOC score and interquartile range (IQR), with the lower and upper whiskers extending to Q1 − 1.5IQR and Q3 + 1.5IQR. Results shown for 23 of the top 30 traits in (A) for which a relevant tissue could be defined. (C) Input CVGs and RVGs for glucose measurement, displayed as a subnetwork derived from PCNet 2.0. Non-prioritized genes are input CVGs or RVGs that did not meet the inclusion criteria for the glucose measurement trait-specific network. (D) KEGG Pathway (top) and Human Phenotype Ontology (bottom) enrichments for the glucose measurement trait-specific network ($n = 670$ genes). The enrichments were filtered to those with an adjusted $p$-value < $5 \times 10^{-5}$, a term size <1000, and a gene intersection size >20, and were clustered based on Jaccard similarity of annotated genes using the Nearest Point Algorithm. Source data are available online for this figure.

Examining the glucose measurement network formed from the convergence of CVGs and RVGs, the ubiquitously expressed CVGs and pancreas-specific RVGs converged to crucial processes underlying glucose regulation (Fig. 5D), such as AMPK signaling ($p_{adj} = 3.5 \times 10^{-29}$), glucagon signaling ($p_{adj} = 2.2 \times 10^{-28}$), insulin signaling ($p_{adj} = 1.5 \times 10^{-25}$), and insulin resistance ($p_{adj} = 7.5 \times 10^{-24}$). The glucose measurement network was also enriched for relevant human phenotypes such as an increased waist-to-hip ratio ($p_{adj} = 7.5 \times 10^{-20}$) and Type II Diabetes Mellitus ($p_{adj} = 1.4 \times 10^{-8}$). Therefore, integrating the glucose measurement associations within a biological network, rather than focusing solely on the CVGs and RVGs, prioritizes the most relevant genes and more fully captures the underlying biological processes.

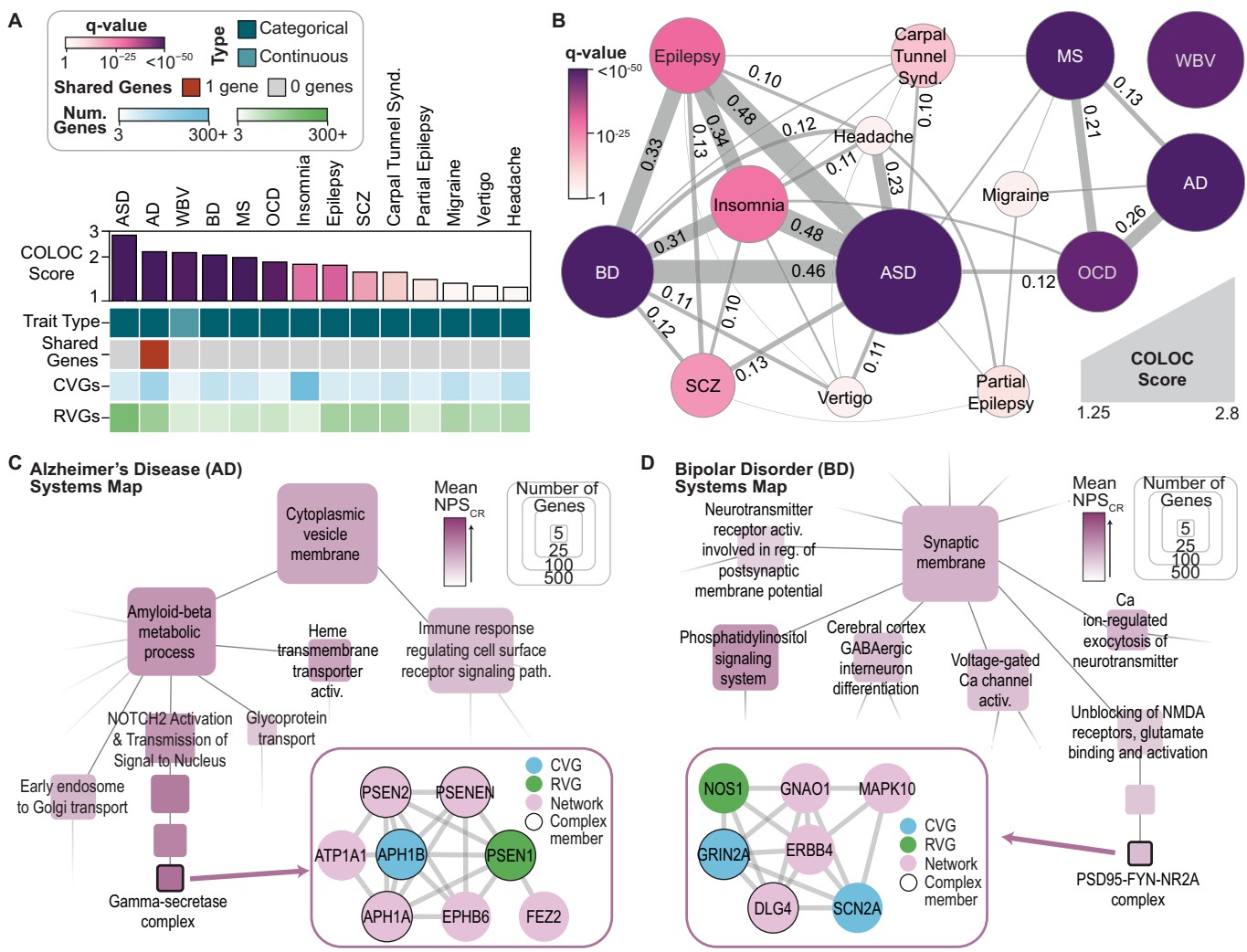

**Figure 6. Network convergence of common and rare variant associations for neuropsychiatric traits.**

(A) Network colocalization results and trait features for the 14 neuropsychiatric traits with significant convergence of common and rare variant-associated genes. (B) Cosine similarity network of 14 neuropsychiatric traits. Edge weights and labels show the cosine similarities of the $NPS_{CR}$ for the union of genes in the individual trait-specific networks. Edges shown for similarities ≥ 0.05 and edges labeled for similarities ≥ 0.10. Node size and color show the COLOC score and associated COLOC q-value, respectively. (C, D) Subsets of the systems maps for Alzheimer's disease (C) and bipolar disorder (D), generated via hierarchical community detection of the trait-specific networks and annotated using g:Profiler (Methods). Node colors indicate the mean $NPS_{CR}$ across all system genes, and node size indicates the number of system genes. Insets show the subnetwork of PCNet 2.0 corresponding to the selected systems, with node colors reflecting the gene status as input associations (CVG, RVG) or prioritized by the network analysis (Network). Underlying trait-specific networks were filtered to genes expressed in at least one brain region with TPM > 1 (HPA). See also Fig. EV5 for full systems maps. Data information: ASD: autism spectrum disorder, AD: Alzheimer's disease, BD: bipolar disorder, MS: multiple sclerosis, OCD: obsessive-compulsive disorder, SCZ: schizophrenia, WBV: whole brain volume. Source data are available online for this figure.

## Systems impacted by common and rare variants associated with neuropsychiatric traits

Using neuropsychiatric traits as an example, we examined the mechanisms underlying the concordance of common and rare variant associations. Among the 21 neuropsychiatric traits, 14 showed significant network convergence ($q_{COLOC} < 0.05$, Fig. 6A). Autism spectrum disorder (ASD) had the strongest convergence (COLOC score = 2.8), consistent with the high genetic heritability (Sandin et al, 2017) and the known contributions of both common and rare variants to this condition (Ben-David and Shifman, 2012). Given the reported genetic similarities across neuropsychiatric traits

(Brainstorm Consortium et al, 2018), we assessed the similarity of the trait-specific networks using the cosine similarity of network proximity scores for the union of the trait-specific network genes. These similarities revealed two distinct clusters (Fig. 6B). The larger cluster included ASD, bipolar disorder (BD), epilepsy, insomnia, schizophrenia (SCZ), and headache, and the smaller cluster included Alzheimer's disease (AD), multiple sclerosis (MS), and obsessive-compulsive disorder (OCD). The greatest similarity was observed between ASD and insomnia, consistent with the high co-occurrence of these disorders (Souders et al, 2017).

Next, we generated hierarchical systems maps to explore multi-scale relationships in the trait-specific networks for the three traits

with the most significant convergence: AD, BD, and ASD. For AD, CVGs and RVGs converged on two major biological processes involved in AD pathogenesis (Figs. 6C and EV5A): one related to amyloid-beta metabolism ($p = 6.2 \times 10^{-21}$) and the other to immune signaling ($p = 1.1 \times 10^{-37}$). One protein complex impacted by both CVs and RVs was the gamma-secretase complex, which catalyzes the proteolysis of the amyloid-beta precursor. RVs in the subunit gene presenilin 1 (PSEN1) were significantly associated with AD ($p_{RV} = 3 \times 10^{-7}$) (Fiziev et al, 2023), as was the CV rs117618017 ($p_{CV} = 4 \times 10^{-20}$) (Lake et al, 2023), which falls within the gamma-secretase subunit gene APH1B. Therefore, despite implicating different genes within the complex, both CVs and RVs may impact AD via modulation of the gamma-secretase complex. For BD, CVGs and RVGs converged to a range of synaptic and neuronal signaling processes (Figs. 6D and EV5B), including phosphatidylinositol signaling ($p = 1.2 \times 10^{-31}$), GABAergic interneuron differentiation ($p = 1.3 \times 10^{-5}$), and regulation of neurotransmitters such as glutamate ($p = 9.4 \times 10^{-12}$). Within glutamate signaling, CVs were identified in the glutamate ionotropic receptor subunit GRIN2A ($p_{CV} = 9 \times 10^{-11}$) (Li et al, 2021a) and the sodium voltage-gated channel subunit SCN2A ($p_{CV} = 2 \times 10^{-9}$) (Li et al, 2021a), alongside RVs in nitric oxide synthase 1 (NOS1, $p_{RV} = 4.1 \times 10^{-7}$) (Lescai et al, 2017). In concert with other interacting proteins, perturbations to these proteins may collectively influence BD via impacts on the excitatory synapse.

Given the high network similarity observed between BD and ASD (Fig. 6B), we compared the functions identified by convergent CVGs and RVGs in each disorder (Fig. EV5B,C). For both BD and ASD, the CVGs and RVGs converged broadly on synapse components. However, variants associated with ASD specifically implicated synapse organization ($p = 4.1 \times 10^{-107}$) and developmental processes such as netrin signaling ($p = 1.4 \times 10^{-11}$). On the other hand, variants associated with BD implicated systems related to metabolic compounds, such as methionine ($p = 6.5 \times 10^{-5}$) and high-density lipoprotein ($p = 7.5 \times 10^{-6}$). These findings underscored the shared neuronal mechanisms of BD and ASD, while also identifying distinct processes impacted by common and rare variants in each disorder.

## Discussion

Even as association studies have grown markedly in population size, the degree to which common variants (CVs) and rare variants (RVs) mediate phenotypes via shared effector genes and biological processes remains unclear. Across 373 diverse human traits, we have demonstrated that CVs and RVs indeed do converge on shared biological networks for most complex traits (76%, Figs. 2 and 3). Given that only a quarter of associated CVs and RVs impacted shared effector genes (Fig. 1), this widespread network convergence indicates that studies of common and rare variants frequently act on complementary genes within shared biological systems.

Each genetic study design likely detects a subsample of the complete set of causal genes, influenced by varying biological and technical factors. Consistent with this premise, we observe significant differences between CV-associated genes (CVGs) and RV-associated genes (RVGs) in terms of structure, expression, and mutational constraint (Fig. 1C). These differences likely drive the

low concordance of implicated genes. While some studies report the coincidence of RVs at GWAS loci as high as 70–97% (preprint: Hawkes et al, 2026; Fiziev et al, 2023; Kim et al, 2022b; Zhou et al, 2023b; preprint: Balderson et al, 2025), others have reported that only 26% of RVs are contained within top GWAS loci (Spence et al, 2026) with low concordance in the implicated genes (Fiziev et al, 2023; Gazal et al, 2022; Spence et al, 2026). Recently, Zhou et al defined the COmmon variant and RAre variant Convergence (CORAC) signature to assess the gene-level overlap of CVGs and RVGs, finding higher CORAC estimates for continuous traits with large effective sample sizes. In comparison, traits with high polygenicity and stronger negative selection produced lower CORAC estimates (Zhou et al, 2023a). Taken together, these findings suggest that the observed overlap of CVGs and RVGs is limited by incomplete study power, the complexity of genetic architectures, and the influences of selection on observed variants.

Under these conditions, looking beyond individual loci and genes is essential for understanding the joint role of CVs and RVs in complex traits. Wiener et al showed that CVs and RVs implicate similar cell types and have pleiotropic effects on similar traits, suggesting mechanistic convergence of variants at a systems level (Weiner et al, 2023). We build on these results by integrating CVs and RVs within a comprehensive biological network to enable both quantification and interpretation of the systems-level convergence (Fig. 2A). Across a range of biological contexts, networks have aided in identifying convergent complexes and pathways from disparate genetic data, including for the study of CVs and RVs in select traits (Chang et al, 2018; Ben-David and Shifman, 2012). Our study substantially expands the scope of these applications to assess network convergence of variants across a diverse range of human traits, including association studies performed in various human populations. In doing so, we confirm the convergence of disparate variants to consistent underlying processes for the majority of traits (Fig. 2) and identify factors driving variability in this convergence (Fig. 4).

A key advantage of our network approach lies in the generation of trait-specific networks from CVGs and RVGs, which allow multi-scale interrogation of how genetic variation converges on biological mechanisms. For example, at a broad biological process or molecular pathway level, genetic variants for Alzheimer's disease (AD) were partitioned into two facets: amyloid-beta metabolism and immune signaling (Figs. 6C and EV5A). At the more granular level of individual protein complexes, we identified a convergence of AD-associated variants on gamma-secretase, a protease complex that has been studied as a potential drug target for this disease (Hur, 2022). In bipolar disorder (BD) and autism spectrum disorder (ASD), genetic variants converged on synaptic structure and function (Figs. 6D and EV5B,C), but diverged in other processes such as methionine metabolism (Carney et al, 1989). These examples illustrate how common and rare variants, even when located in different genetic loci, implicate similar functions across multiple levels of biological organization. However, additional studies are needed to identify the causal mechanisms underlying the variant associations and validate the role of these convergent systems in disease.

Across the 373 traits studied, we observed greater concordance for traits that are highly heritable, quantitative, or involve more mutationally constrained genes (Fig. 4). The finding that traits with high SNP heritability exhibit stronger network convergence

(Fig. 4A) builds on previous findings that common and rare variant heritability are correlated (Weiner et al, 2023) and that incorporating rare variants enhances heritability prediction (Wainschtein et al, 2022). Using the CORAC signature, greater convergence of highly powered quantitative traits was similarly observed (Zhou et al, 2023a), but highly polygenic traits had dampened convergence. In contrast, we found that traits with more numerous association signals tended to show stronger network convergence (Fig. EV3). While factors such as study power and design confound the number of genes identified, this result highlights the advantage of a network approach for capturing convergence in complex genetic architectures.

We also observed that traits with RVs in tissue-specific genes showed stronger CVG/RVG convergence (Figs. 4H and 5), suggesting that the focused expression of these genes may help define the core structure of a trait's genetic architecture. Results from RV burden tests reinforce this interpretation, as they tend to identify genes that are more trait-specific (Spence et al, 2026). In this study, we utilized bulk-tissue expression to capture broad patterns of gene activity, meaning that the observed enrichments may reflect aggregate effects across multiple cell types rather than convergence within specific cell populations. Indeed, differences in the specificity of regulatory effects between variant classes have been reported: Li et al reported that RV expression quantitative trait loci (eQTLs) are more tissue-specific than CV eQTLs (Li et al, 2021b), whereas Cuomo et al found that CV eQTLs are more cell-type-specific than RV eQTLs (preprint: Cuomo et al, 2025). Future integration of eQTLs and single-cell expression data alongside common and rare variants could clarify the roles of tissue-specific and cell-type-specific variants.

Overall, the extent of convergence between common and rare variants was consistent across different association studies and network resources (Figs. 3B and EV2C,D). However, for some traits, the selection of input CV and RV studies influenced the observed convergence (Figs. 3C and EV2A). Such discrepancies likely result from differences in study power, population cohorts, confounding variables, or trait definitions. While we focused on study pairs with the strongest convergence, further analysis of cases where convergence depends on the cohort composition could prove informative for translational applications. Similarly, some traits showed variable results with different networks, possibly due to the varying representation of systems and datatypes within each resource (Wright et al, 2025). Network convergence could also be evaluated using NetColoc with targeted network resources, such as regulatory or signaling-specific networks (Lo Surdo et al, 2022; Liska et al, 2022; Wright et al, 2025). Implementing NetColoc with these resources will require additional adaptation and benchmarking to leverage the directed biological relationships present in some networks. Ultimately, we anticipate that such applications will elucidate mechanism-specific subsets of convergent genetic variation.

Several limitations should be considered when interpreting these results. Some distinct properties of CVGs and RVGs may result from underlying selection biases and limitations of the SNP-to-Gene mapping procedure. For example, larger genes have more opportunities to be tagged by a causal variant in GWAS, which is reflected in our finding that CVGs tended to be longer (Fig. 1C). This bias may be amplified via our positional SNP-to-Gene mapping (Jia et al, 2010). While positional mapping is foundational

for identifying causal genes from CVs (Gazal et al, 2022), it does not capture the full complexity of gene regulation. To maximize the accuracy of positional SNP-to-Gene mapping, we chose to only examine CVs that fell directly within gene regions. However, excluding intergenic variants may have excluded a non-random subset of functional variants, which could have skewed the observed convergence results. Future studies could leverage refined methods for SNP-to-Gene mapping, such as those utilizing linkage disequilibrium (de Leeuw et al, 2015), to improve the accuracy of input genes. Lastly, while our results demonstrate the mechanistic convergence of common and rare variants, this does not imply their co-occurrence within individuals. Understanding the correlation or mutual exclusivity of convergent variants and systems across populations could inform the stratification of individuals based on overall genetic risk and underlying genetic architecture.

In conclusion, by harmonizing studies of common and rare variants with a comprehensive biological knowledge network, we find that these association studies yield complementary genes that converge on consistent biological systems. This study provides a foundation for systematically understanding the shared roles of diverse genetic signals within complex traits and defines trait-specific networks to accelerate further research. Future applications of this framework could incorporate additional variant and gene types, such as intergenic variants, de novo mutations, and copy number variants, to achieve an even more complete understanding of the genetic etiology of human traits.

## Methods

**Reagents and tools table**

| Reagent/Resource | Reference or Source | Identifier or Catalog Number |
|---|---|---|
| **Experimental models** | | |
| N/A | | |
| **Recombinant DNA** | | |
| N/A | | |
| **Antibodies** | | |
| N/A | | |
| **Oligonucleotides and other sequence-based reagents** | | |
| N/A | | |
| **Chemicals, Enzymes and other reagents** | | |
| N/A | | |
| **Software** | | |
| Python 3.10 | https://www.python.org/ | |
| NetColoc 1.1.0a1 | https://pypi.org/project/netcoloc/ (Rosenthal et al, 2023) | |
| neteval 0.2.3a1 | https://github.com/sarah-n-wright/Network_Evaluation_Tools (Wright et al, 2025) | |
| Sentence_transformers 3.4.1 | https://sbert.net/ (preprint: Reimers and Gurevych, 2019) | |
| Obonet 1.0.0 | https://github.com/dhimmel/obonet | |

| Reagent/Resource | Reference or Source | Identifier or Catalog Number |
|---|---|---|
| Goatools 1.5.2 | https://github.com/tanghaibao/goatools (Klopfenstein et al, 2018) | |
| Scikit-learn 1.5.1 | https://scikit-learn.org/1.5/index.html | |
| Networkx 2.8.8 | https://networkx.org/ (Hagberg et al, 2008) | |
| NDEx Python Client 3.9.0 | https://pypi.org/project/ndex2/ (Pillich et al, 2021) | |
| Omnipath 1.0.8 | https://pypi.org/project/omnipath/1.0.8/ (Türei et al, 2021) | |
| R 4.2 | https://cran.r-project.org/ | |
| TissueEnrich 1.18.0 | https://www.bioconductor.org/packages/3.16/bioc/html/TissueEnrich.html (Jain and Tuteja, 2019) | |
| G:Profiler | https://biit.cs.ut.ee/gprofiler/gost (Kolberg et al, 2023) | |
| Adobe Illustrator CC 2024 | https://www.adobe.com/ | |
| Cytoscape 3.10.3 | https://cytoscape.org/ (Shannon et al, 2003) | |
| Cytoscape Web 1.0.4 | https://web.cytoscape.org/ (Ono et al, 2025) | |
| CyCommunityDetection 1.12.1 | https://apps.cytoscape.org/apps/cycommunitydetection (Singhal et al, 2020) | |
| MAGMA 1.10 | https://cncr.nl/research/magma/ (de Leeuw et al, 2015) | |
| **Other** | | |
| N/A | | |

## Data curation

### Biological knowledge networks

The PCNet, STRING and HumanNet networks were sourced from the Network Data Exchange (NDEx, ndexbio.org).

- PCNet 2.0 (https://doi.org/10.18119/N9JP5J) contains all interactions present in at least two of the top 15 high-performing interaction networks (Wright et al, 2025).
- PCNet 2.2 (https://doi.org/10.18119/N9960N) is a reduced version of PCNet 2.0 that includes all interactions present in at least two of the top ten high-performing interaction networks, without co-citation evidence (Wright et al, 2025).
- We sourced standardized versions of the STRING 12.0 (Szklarczyk et al, 2023) network and the HumanNet v3 XC (Kim et al, 2022a) network from the 'State of the Interactomes - Source Networks' network set (https://doi.org/10.18119/N95C9J).
- The STRING network was filtered to interactions with a score >700 to define the STRING High Confidence (HC) network.

We defined the OmniPath network using the omnipath Python client (Türei et al, 2021), sourcing all interactions except those

involving small molecules. We binarized all multiplex interactions and standardized the networks using the Python package neteval v0.2.3a1 (Wright et al, 2025) to remove duplicates and convert identifiers to NCBI Gene IDs. All interactions were treated as undirected.

### Gene identifier mapping

Gene identifiers were mapped using the gene_mapper module from neteval v0.2.3a1 (Wright et al, 2025). Input identifiers were first updated to the most recent versions in the original database. Unless otherwise specified, all gene identifiers were then converted to NCBI Gene IDs.

### Trait semantic similarity

Trait identifiers and descriptions were sourced from the Experimental Factor Ontology (EFO) (Malone et al, 2010) (http://www.ebi.ac.uk/efo/efo.obo) via obonet v1.0.0 (https://pypi.org/project/obonet/). To assess the similarity of trait descriptions, we utilized trait embeddings generated with the pretrained Sentence-BERT (SBERT) via the sentence_transformers Python package (preprint: Reimers and Gurevych, 2019), as follows:

1. Expand common acronyms (e.g., HDL: high-density lipoprotein).
2. Embed all expanded trait descriptions using the model 'all-MiniLM-L6-v2'.
3. Calculate the cosine similarity between two trait embeddings to quantify the similarity of trait descriptions.

### Common variant associations

Results from common-variant genome-wide association studies (GWAS) were downloaded from the GWAS Catalog on January 29, 2025 (Data ref: GWAS Catalog Associations, 2025), along with accompanying trait (Data ref: GWAS Catalog Trait Mappings, 2025) and study (Data ref: GWAS Catalog Study Information, 2025) metadata. We defined sets of common-variant-associated genes (CVGs) using the following procedure:

1. Take gene-level associations from the MAPPED_GENE column provided by the GWAS Catalog. Briefly, a SNP is mapped to a gene if it falls within the gene region, as defined by the Ensembl location (Harrison et al, 2024).
2. Filter associations to those falling within a single gene region to define a higher confidence set of associated genes. Where multiple SNP associations mapped to a single gene, the strongest $p$-value is maintained for the gene association.
3. Define set of common-variant-associated genes (CVGs) for each trait and study as the set of distinct genes tagged by SNPs with $p < 1 \times 10^{-5}$'.
4. Filter individual studies based on study metadata. We excluded studies were if they: (a) listed a background trait ('MAPPED BACKGROUND TRAIT'), (b) utilized targeted sequencing or exome sequencing and therefore did not represent genome-wide analyses ('GENOTYPING TECHNOLOGY'), (c) did not list study population size ('INITIAL SAMPLE SIZE'), or (d) reported fewer than three distinct gene associations with $p < 1 \times 10^{-5}$ present in PCNet 2.0.

Mappings between reported traits ('DISEASE/TRAIT') and Experimental Factor Ontology (EFO) terms

('MAPPED_TRAIT_URI') in the GWAS Catalog are not one-to-one. Therefore, we standardized EFO trait mappings by:

1. Maintaining all one-to-one mappings.
2. Excluding studies listing multiple EFO terms for a single reported trait.
3. Taking the EFO terms with the highest semantic similarity for the same reported trait mapped to multiple EFO terms across studies.

### Rare variant associations

We sourced rare variant gene-level association results and related metadata from the Rare Variant Association Repository (RAVAR) database (Cao et al, 2024; Data ref: RAVAR Associations, 2024) on June 11, 2024. Additional metadata, including population size, study design, and population cohort, were manually curated from the associated publications for the included studies. Where available, we took the population sample size for individual traits within multi-trait studies. The set of rare-variant-associated genes (RVGs) for each trait was defined as the set of distinct genes with gene-level association $p < 1 \times 10^{-4}$. Studies were excluded if:

- Associations were collated from literature reviews or sequencing studies that were not genome-wide or exome-wide.
- The population size could not be determined.
- Fewer than three distinct gene associations with $p < 1 \times 10^{-4}$ were reported for genes present in PCNet 2.0.

Mappings of reported trait descriptions ('Reported Trait') and EFO terms were harmonized with the GWAS catalog, using the same procedure described for the common variant studies. If a reported trait did not have an exact match in the GWAS catalog, the EFO mapping supplied by RAVAR ('Trait Ontology id') was maintained. This process ensured consistent EFO mappings between the GWAS Catalog and RAVAR.

### Common and rare trait & study matching

We identified traits with matched common and rare variant results using the mapped EFO terms. For traits with multiple independent studies, we prioritized a single study for the initial analysis and up to four additional studies for the replication analyses.

The initial common variant study per trait was selected as the GWAS with the largest population sample size, after filtering for high trait similarity between the reported trait description ('DISEASE/TRAIT') and the mapped EFO term description, and ensuring the availability of cohort information and summary statistics. The initial rare variant study per trait was selected as the study with the largest population sample size, after filtering for high trait similarity between the reported trait description ('Reported Trait') and the mapped EFO term description, and the availability of cohort information. After matching based on EFO terms, the trait semantic similarities of the original trait descriptions (GWAS Catalog: 'DISEASE/TRAIT', RAVAR: 'Reported Trait') were calculated. All study combinations with a similarity of less than 0.7 were assessed manually, and only those that represented synonyms of the same trait were maintained for the analysis.

Up to four additional GWAS and four additional rare variant studies were selected per trait for replication analysis. Studies were

required to have a cosine similarity between the reported trait and the mapped EFO term of at least 0.7. Where possible, studies utilizing different population cohorts were prioritized, with further prioritization based on the population sample size.

Information on all selected studies and accompanying gene sets can be found in Dataset EV1. For each combination of common and rare studies within each trait, the gene-level convergence was calculated by a hypergeometric test between the sets of CVGs and RVGs. A background of 19,000 was used to approximate the total number of human protein-coding genes.

### Trait and study properties

Traits were manually classified as categorical if they involved a case-control study or other discrete phenotype, or continuous if they involved a quantitative measurement. All traits were manually assigned to high-level biological domains. Domains with fewer than 5 traits in the analysis set were assigned to the 'Other' category for visualization purposes.

Trait SNP heritability estimates were sourced from the Neale Lab Round 2 analysis (Data ref: UKBB, 2022), calculated via partitioned LD Score regression (LDSC) for 4236 phenotypes in the UK Biobank. In total, we identified heritability estimates for 197 traits.

- We excluded ten pigmentation and bilirubin phenotypes based on their unusually high standard error (SE) and low polygenicity, as flagged in the UKB heritability analysis.
- We also excluded three traits flagged as having ordinal coding issues based on the column 'isBadOrdinal'.
- For each of our 373 traits, we identified the most semantically similar trait in the UKB heritability results and manually curated the matches to remove duplicates and imprecise matches.

Disease prevalence estimates were sourced from the Global Burden of Disease (GBD) study from the Institute for Health Metrics and Evaluation (Data ref: GBD, 2021). In total, we identified prevalence estimates for 87 traits.

1. We extracted disease prevalence as a percent of the population for the five years from 2017 to 2021, including all locations, ages, and sexes.
2. We took the mean prevalence across the five-year period.
3. For each of our 373 traits, we identified the most semantically similar trait in the GBD prevalence results and manually curated the matches to remove duplicates and imprecise matches.

### Gene properties and annotations

Gene region annotations were sourced from Ensembl v113 (Harrison et al, 2024), downloaded on February 14, 2025. Entries were filtered to transcripts on hg38 primary chromosome assemblies with a mapped HGNC Symbol. Gene length was calculated as the difference between the start (bp) and end (bp) positions. Genes with conflicting entries were excluded.

Median gene-level mRNA expression by tissue was sourced from GTEx v8 (GTEx Consortium et al, 2017; Data ref: GTEx Portal, 2017).

- For genes with multiple transcripts, we consolidated the values for all transcripts within each tissue. We took the mean of all transcripts if all values had a similar magnitude (all TPM observations above or below $10^{-4}$); otherwise, we took the maximum.

- Overall expression levels were calculated as the mean across all tissues. Where a gene had multiple transcripts, the transcript with the highest mean expression was selected.
- The number of expressed tissues per gene was defined as the number of tissues with TPM > 1.

Functional Gene Ontology (GO) associations were downloaded from NCBI using goatools (Klopfenstein et al, 2018) on November 25, 2024.

- Entries with the negating qualifier 'NOT' were excluded.
- We extracted the number of distinct GO IDs associated with each human NCBI Gene ID across the three branches of GO: Cellular Component (CC), Biological Process (BP), and Molecular Function (MF).
- Missing genes were assigned a value of zero.

Gene selective constraint scores, as measured by $s_{het}$ (an estimate of the fitness reduction for heterozygous carriers of an LOF mutation in a gene), were sourced from Zenodo on August 7, 2025 (Zeng et al, 2024; Data ref: GeneBayes, 2023). We took the posterior mean value of $s_{het}$ for each gene.

Gene mutational constraint estimates based on loss-of-function (LoF) mutations were sourced from gnomAD v4.1 (Karczewski et al, 2020; Data ref: gnomAD, 2024). For each gene, we extracted the LOEUF (loss-of-function observed/expected upper bound fraction) (Lek et al, 2016; Karczewski et al, 2020), the synonymous intolerance Z-score (Samocha et al, 2014; Lek et al, 2016), and the expected number of LoF mutations. The LOEUF metric provides a continuous metric of a gene's intolerance to loss-of-function variants, with a threshold of LOEUF < 0.6 recommended for defining LoF-constrained genes. For each distinct gene symbol, we prioritized entries associated with canonical and MANE Select transcripts, followed by those with an assigned NCBI Gene ID.

The lengths of coding sequences for each gene (CDS) were derived from GENCODE v46 Basic Gene Annotation (Frankish et al, 2019; Data ref: GENCODE, 2024). Positional gene conservation scores (phyloP) were sourced from the UCSC Genome Browser (Nassar et al, 2023; Data ref: UCSC Genome Browser, 2017), calculated via multiple alignment of 29 vertebrate species to the hg38 human genome. PhyloP scores from the reference chromosomes were aggregated for each transcript using BEDOPS v2.4.41 (Neph et al, 2012). The final gene conservation score was defined per gene as the mean phyloP across all positions within the gene's CDS.

Summary values were calculated for each set of RVGs and CVGs by taking the median value for all genes in the gene set. Genes with missing values were excluded. Paired differences between common and rare gene sets for matched traits were assessed by the Wilcoxon Signed-Rank test with BH correction.

## Quantitative network colocalization (Q-NetColoc)

### Network proximity scores (NPS)

For network propagation and colocalization, we used the Python package NetColoc v1.1.0a1 (https://pypi.org/project/netcoloc/1.1.0a1/). First, sets of common and rare associated genes (CVGs and RVGs) were propagated through the network using a Random Walk with Restart process (Vanunu et al, 2010). For efficient

computation, we utilized the closed-form solution for the random walk process described by Eq. 1:

$$F = (I - \alpha W)^{-1}(1 - \alpha)Y_0 \qquad (1)$$

Where $F$ is the stable heat vector for all nodes, $Y_0$ is the initial heat for all nodes, $W$ is the column-normalized adjacency matrix of the underlying network, and $\alpha \in (0,1)$ is the dissipation constant. For all analyses, we used the default value of $\alpha = 0.5$. In the original NetColoc method, the initial heat vector $Y_0$ is assigned 1 for all genes in the input set, and 0 otherwise. We refer to this as binary NetColoc (B-NetColoc). Given the lenient thresholds used to define CVGs and RVGs, we modified NetColoc to weight all CVGs and RVGs based on the negative-$\log_{10}$ association $p$-value to appropriately up-weight the most confident associations. Therefore, we defined quantitative NetColoc (Q-NetColoc), where:

- $Y_0$ is assigned the value of $-\log_{10}(p)$ for each associated gene with $p < \theta$, and 0 otherwise, where $\theta$ is the significance threshold.
- The values in $Y_0$ are normalized such that the sum of all values is 1.

Following network propagation, Network Proximity Scores (NPS) were calculated for each gene in the network as a Z-score of the observed heat relative to a background null distribution. This null distribution was formed for each gene via network propagation of degree-matched random gene sets:

1. All network genes were binned by network degree to create bins containing a minimum of β genes.
2. For B-NetColoc, we randomly selected gene sets of equal size from the bins corresponding to the input gene set. For Q-NetColoc, we shuffled scores of all genes within each bin in order to maintain the degree-score correlation.
3. For each gene $g$, we then calculated $NPS_{g,S}$ as:

$$NPS_{g,S} = \frac{\log(F_{g,S}) - \langle \log(F_{g,rand}) \rangle}{\sigma(\log(F_{g,rand}))} \qquad (2)$$

Where $F_{g,S}$ is the observed heat at $g$ after propagation of gene set $S$, and $F_{g,rand}$ is the null distribution of heats from randomized gene sets. The mean of a vector is denoted by $\langle \ldots \rangle$ and the standard deviation of a vector is denoted by $\sigma(\ldots)$. All values are log-transformed to ensure the distributions are approximately normally distributed.

4. Following the calculation of $NPS_{g,R}$ and $NPS_{g,C}$ for all genes, we calculated the combined network proximities $NPS_{CR}$ as the product of the independent gene scores:

$$NPS_{CR} = NPS_C \times NPS_R \qquad (3)$$

### Colocalized trait-specific networks and COLOC score

To assess the convergence of CVGs and RVGs within the network, we defined colocalized trait-specific network $G_{trait}$ for each trait as:

$$G_{trait} \in g \, \Bigg| \begin{cases} NPS_{g,C} > \tau \\ NPS_{g,R} > \tau \\ NPS_{g,CR} > \tau^* \end{cases} \qquad (4)$$

Where $\tau$ is the threshold on individual NPS, and $\tau^*$ is the combined NPS threshold. The observed size of each $G_{trait}$ was compared to a permuted null distribution to calculate the network colocalization statistics:

- We permuted the labels of $NPS_C$ 1000 times, each time calculating the number of genes meeting the thresholds for inclusion in the trait-specific network (Eq. 4).
- For genes shared between the common and rare input sets, labels were permuted separately to maintain the higher expected distribution for these genes.
- The significance of network colocalization (COLOC P) was defined by a Z-test comparing the observed size of $G_{trait}$ to the expected size of $G_{trait}$.
- We defined the network colocalization (COLOC) score as the observed over expected size of the trait-specific network.

### Optimization and benchmarking of Q-NetColoc

Our implementation of Q-NetColoc was benchmarked and optimized using GWAS results from the GWAS Catalog for a random sample of traits not included in the main analysis. In addition to the criteria outlined in *Common variant associations*, we selected studies with publicly available summary statistics, a single-ancestry cohort, and a defined genome build. In total, we generated a testing set comprising 87 studies covering 43 distinct traits:

1. From each study, we extracted the common variant-associated genes (CVGs) as described in *Common variant associations*, which we define as the Direct SNP-to-Gene (S2G) approach.
2. All sets of CVGs were partitioned into two non-overlapping sets of equal size.
3. Test pairs were formed by matching two halves of the same study, and control pairs were formed by matching two halves of studies of unrelated traits.
4. For Q-NetColoc, all *p*-values below the association *p*-value threshold ($\theta$) were transformed to negative-log$_{10}$ scores and normalized to have a sum of 1.
5. For B-NetColoc, *p*-values were binarized as significant (1) and non-significant (0) based on the association *p*-value threshold ($\theta$).

Assessment metrics. We assessed the performance of different NetColoc configurations using the Area Under the Receiver Operating Characteristic curve (AUROC). Within each group of configurations below, we subset the results to all pairs evaluated by all configurations and a matching number of control pairs. AUROC was calculated using the roc_auc_score function from scikit-learn v1.5.1 (Pedregosa et al, 2011). AUROC was calculated for the COLOC score and the associated colocalization *p*-value (COLOC P). Full results for the optimization of Q-NetColoc are included in Dataset EV2.

Thresholds. We first assessed the optimal thresholds $\tau$ and $\tau^*$ for defining the trait-specific network for Q-NetColoc and B-NetColoc (Eq. 4).

- Thresholds were scanned in {1.0, 1.5, 2.0, 2.5, 3.0, 3.5, 4.0} for $\tau^* \geq \tau$.
- A default bin size of $\beta = 10$ for degree-matched randomization was used.

The parameter $\tau$ had a greater effect on the observed AUROC than $\tau^*$, with lower values giving better performance for Q-NetColoc and B-NetColoc (Appendix Fig. S2A). Therefore, we proceeded with $\tau = 1$, paired with a stricter value of $\tau^* = 3$, for all subsequent optimization and analysis tasks.

Bin size. The bin size ($\beta$) for performing degree-matched randomization balances the randomization of scores with the maintenance of the degree-score correlation. We tested values of the bin size parameter $\beta$ in {5, 10, 20, 40, 80, 160} for B-NetColoc and Q-NetColoc. Based on the AUROC values, we determined an optimal bin size of $\beta = 20$ (Appendix Fig. S2B).

Association *p*-value. Initial testing was performed using a lenient association *p*-value threshold of $\theta_{lenient} = 1.0 \times 10^{-5}$ for determining CVGs. While this allows for the inclusion of more genes and traits, it also increases the inclusion of false-positive associations. Therefore, we also tested a strict threshold of $\theta_{strict} = 1 \times 10^{-8}$. The lenient threshold consistently provided better separation between test and control pairs (Appendix Fig. S2C). For example, looking at COLOC scores from Q-NetColoc, the lenient threshold gave AUROC = 0.84, compared to AUROC = 0.82 for the strict threshold. Therefore, we determined that the benefits of including the appropriately weighted sub-threshold associations outweighed the potential inclusion of more false positives. Across all tests, minimal differences were observed between Q-NetColoc and B-NetColoc.

SNP-to-Gene mapping. Finally, we compared the default SNP-to-Gene mapping approach (S2G) to the gene aggregation approach MAGMA v1.10 (de Leeuw et al, 2015). For each testing study, we:

1. Standardized the format of the summary statistics.
2. Annotated the SNPs using the assigned genome build.
3. Calculated gene-level *p*-values with MAGMA, using the most relevant population reference from the 1000 Genomes Project (1000 Genomes Project Consortium et al, 2015), based on the listed study ancestry.

MAGMA and all auxiliary files were downloaded from https://cncr.nl/research/magma/ on December 5, 2024. For MAGMA gene sets, we defined the strict threshold as $\theta_{strict} = 2.5 \times 10^{-6}$, corresponding to the typical multiple hypothesis correction for the number of coding genes. We defined the lenient threshold as $\theta_{lenient} = 1.0 \times 10^{-4}$. As with the S2G gene sets, the lenient threshold yielded better results for MAGMA. Overall, MAGMA results provided slightly better separation between test and control pairs than S2G. For example, based on the COLOC score from Q-NetColoc, MAGMA achieved an AUROC = 0.89 using the lenient threshold (Appendix Fig. S2D). Therefore, where practical, MAGMA gene-level association statistics can boost NetColoc performance. However, variability in the structure and availability of summary statistics and population ancestry precluded the use of MAGMA for our primary analysis.

Input gene set dilution testing. To test the response of NetColoc to the introduction of random error in the gene sets, we progressively diluted the input data with degree-matched random genes. All

dilution analyses were performed using the parameters $\beta = 20$, $\tau = 1$, and $\tau^* = 3$. For testing B-NetColoc, we utilized gene sets derived from the Gene Ontology (GO), with human GO gene associations sourced from NCBI using goatools v1.5.2 (Klopfenstein et al, 2018) on July 11, 2024.

1. All genes associated with a specific GO term were defined as associated with all parent GO terms.
2. GO terms with between 200 and 250 associated genes were selected from each ontology branch to give 48 biological process (BP) gene sets, 22 cellular compartment (CC) gene sets, and 16 molecular function (MF) gene sets.
3. All GO gene sets were subsampled to 150 genes and partitioned into two non-overlapping sets, with three repeats.
4. The partitioned sets were assessed via B-NetColoc, with all producing a significant COLOC score.
5. All GO gene sets were then diluted to contain 25%, 50%, 75% and 100% random genes, with three repeats.
6. All diluted partitioned sets were assessed via B-NetColoc.

For testing Q-NetColoc, we used the partitioned GWAS test gene sets defined by S2G mapping for Q-NetColoc optimizations, with a threshold of $\theta = 1 \times 10^{-5}$.

1. First, we selected partitioned GWAS gene sets with a significant COLOC score ($n = 38$).
2. Then, we diluted each set to contain 25%, 50%, 75%, and 100% random genes, with 5 repeats. When replacing a given gene with a degree-matched random gene, the association $p$-value of that gene was given to the random gene.
3. All diluted gene sets were assessed via Q-NetColoc.

We observed that COLOC scores consistently decayed as the input gene sets were diluted (Appendix Fig. S3A,B). At 100% dilution, the mean of the distribution of COLOC scores was not significantly different from the null expectation of COLOC = 1 for GO gene sets ($\mu_{COLOC} = 0.983$, $p_{GO} = 0.055$, 1-sample t-test) or partitioned GWAS gene sets ($\mu_{COLOC} = 1.03$, $p_{GWAS} = 0.36$, 1-sample t-test), indicating that the permuted null distributions used for assessing network colocalization are well calibrated. Full dilution results are included in Dataset EV3.

Input network dilution testing. To understand the importance of network structure, we tested the response of NetColoc to the introduction of random error in the networks.

- We performed degree-preserving network swaps using the function shuffle_networks from neteval v0.2.3a1 (Wright et al, 2025) to create a series of progressively more randomized versions of PCNet 2.0 and STRING-HC with equivalent topologies.
- The extent of randomization (dilution) was measured based on $1 - J$, where J is the Jaccard similarity of edges between the original and shuffled network.
- For PCNet 2.0, we generated three series of nine networks with dilution fractions between 0.18 and 0.8.
- For STRING-HC, we generated three series of nine networks with dilution fractions ranging from 0.18 to 0.95.

We performed Q-NetColoc with $\beta = 20$, $\tau = 1$, and $\tau^* = 3$ for the partitioned GWAS test gene sets defined by S2G mapping, using a threshold of $\theta = 1 \times 10^{-5}$, and filtered the sets to those with significant colocalization using the original network. We observed that COLOC scores consistently decayed as the network structures were randomized, indicating that the NetColoc procedure depends on the biological knowledge encoded in the edges, rather than simply the network topology (Appendix Fig. S3C,D). Full dilution results are included in Dataset EV3.

### Network colocalization of matched CVGs and RVGs using Q-NetColoc

For the main analysis, Q-NetColoc was implemented with the following parameters:

- Bin size $\beta = 20$.
- Trait-specific network thresholds of $\tau = 1$ and $\tau^* = 3$.
- PCNet 2.0 as the underlying network.
- Input genes weighted based on the negative $\log_{10}$ association $p$-value, with association $p$-value thresholds of $\theta_{CVG} = 1 \times 10^{-5}$ for CVGs and $\theta_{RVG} = 1 \times 10^{-4}$ for RVGs.

We examined the null distribution of expected trait-specific network size (used to determine the colocalization $p$-value) for a subset of test pairs with varying COLOC scores using quantile–quantile plots. This analysis confirmed that the null distributions of trait-specific network size approximated standard normal distributions (Appendix Fig. S3E). The initial network colocalization was performed using the top common variant study and the top rare variant study per trait, as defined in *Common and rare trait & study matching*. The additional analyses were then performed with all pair-wise combinations of common variant and rare variant studies per trait. To avoid redundant calculations, **NPS$_C$** and **NPS$_R$** were calculated once for each set of CVGs or RVGs and reused across different study combinations. Multiple hypothesis correction was performed on the network colocalization $p$-values using a BH correction. Prioritized input genes were defined as input CVGs and RVGs meeting the criteria for inclusion in the trait-specific network (Eq. 4). For each trait, the optimal combination of common variant study and rare variant study was defined as the study combination achieving the highest COLOC score. Using the 373 optimal study combinations, Q-NetColoc was implemented using PCNet 2.2, HumanNet v3, STRING-HC, and OmniPath as the underlying networks, with all other parameters equal. Results were generated for each network for all traits with at least three CVGs and at least three RVGs in that network.

## Analysis and interpretation of common-rare trait-specific networks

### Gene set features

We collected study design features and summary features of the genes in each set of CVGs (C) and RVGs (R) from all analyzed pairs of initial and additional studies ($n = 1634$ total study pairs). For each study pair, we compiled the study population sizes ($N_C$, $N_R$) and the number of genes in the sets ($g_C$, $g_R$).

For each paired set of CVGs and RVGs, we calculated the Jaccard similarity ($J_{CR}$) and identified the subnetworks of PCNet 2.0 induced by the CVGs and induced by the RVGs. We calculated the subnetwork densities ($\rho_C$ and $\rho_R$) based on the number of edges between CVGs and between RVGs, respectively. From the subnetwork of PCNet 2.0 induced by the union of CVGs and RVGs, we

calculated the assortativity of CVGs and RVGs using the function attribute_assortativity_coefficient from networkx v2.8.8 (Hagberg et al, 2008). For gene set pairs with $J_{RC} > 0$, we took the mean assortativity of all common genes to disjoint rare genes and all rare genes to disjoint common genes: $A_{CR} = (A_{C,R-C} + A_{R,C-R})/2$.

For each of the gene properties selective constraint (Constraint), gene size (GeneSize), mRNA expression (mRNA), number of expressed tissues (nTissues) and number of GO annotations (GO) described in *Data curation > Gene properties and annotations*, we calculated the weighted mean value $\mu_{CR} = (g_C\mu_C + g_R\mu_R)/(g_C + g_R)$ across CVGs and RVGs and the normalized difference in means ($\delta$) defined as $\delta = 2\ (\mu_R - \mu_C)/(\mu_R + \mu_C)$ (Fig. EV4B).

We transformed and normalized all features using scikit-learn v1.5.1 (Pedregosa et al, 2011) based on the observed distributions:

- Features $N_C$, $N_R$, $A_{CR}$, $\mu_{Constraint}$, $\mu_{GeneSize}$, $\mu_{mRNA}$, $\mu_{GO}$, $\delta_{Constraint}$, $\delta_{GeneSize}$, $\delta_{mRNA}$, $\delta_{GO}$, $\delta_{nTissues}$ were quantile-normalized to a Gaussian distribution using QuantileTransformer with the number of quantiles set to the sample size.
- The features $g_C$, $g_R$, $\rho_C$, $\rho_R$, $J_{CR}$, $\mu_{nTissues}$ were power-transformed using PowerTransformer.
- We added a one-hot encoded feature to indicate categorical traits ('binary') and a one-hot encoded feature to identify the study pairs with $J_{CR} = 0$ ('jaccard_zero').

To ensure comparable distributions for all features, we then rescaled all features using StandardScaler.

### Elastic Net regression analysis

To estimate the contribution of each feature to the observed network colocalization (COLOC score), we performed a regression analysis using Elastic Net. This model was chosen to balance feature selection and account for correlations between the input features (Fig. EV4A). The target variable, COLOC score, was quantile-normalized (QuantileTransformer) with the number of quantiles set to the sample size. To determine the optimal penalty parameter ($\alpha$) and L1-ratio (l1_ratio), we performed a five-fold cross-validation analysis to select the parameters minimizing out-of-fold mean-squared error (MSE), using ElasticNetCV from scikit-learn. We utilized a log-spaced grid of 20 $\alpha$ values between $10^{-5}$ and $10^1$, and l1_ratios in {0.1, 0.25, 0.4, 0.5, 0.6, 0.7, 0.8, 0.9, 0.95, 0.99, 1}, with all other parameters set to their defaults. Following cross-validation, we identified an optimal $\alpha$ of 0.014 and an l1_ratio of 0.1.

The variance explained by the optimized model was calculated using explained_variance_score from scikit-learn. The optimized model was used to calculate the residual COLOC score as the difference between the predicted and actual COLOC scores. We then performed bootstrapping to estimate the confidence of the coefficient fits. We generated 2000 bootstrap replicates using sampling with replacement and fit an Elastic Net model to each replicate. Coefficients for each feature are reported as bootstrap means with corresponding 95% confidence intervals. Empirical *p*-values were calculated as:

$$p = 2 * \min\left(\frac{r_+ + 1}{2001}, \frac{r_- + 1}{2001}\right) \quad (5)$$

Where $r_+$ is the number of coefficient estimates $>0$ and $r_-$ is the number of coefficient estimates $<0$, followed by Bonferroni correction with the number of features.

### Enrichment for tissue-specific genes

Tissue-specific enrichment analysis (TSEA) was performed using TissueEnrich (Jain and Tuteja, 2019), with all genes mapped to Ensembl gene identifiers. Tissue-specific genes were defined from RNA-seq results from the Human Protein Atlas (HPA (Uhlén et al, 2015)) included in TissueEnrich using default parameters of fold change $\geq 5$, minimum expression $\geq 1$, and number of expressed tissues $\leq 7$. For enrichment analysis, all genes classified as Tissue-Enriched, Tissue-Enhanced, or Group-Enriched were considered. The set of all genes in PCNet 2.0, converted to Ensembl gene identifiers, was used as the background gene list, and correction for multiple hypothesis testing was performed using the BH correction.

First, for all traits with significant network colocalization, we calculated the enrichment of the trait-specific networks for genes expressed in trait-relevant tissues (Fig. 3F). For each trait, we defined four gene sets:

- All genes in the trait-specific network.
- Prioritized genes (input CVGs and RVGs in the trait-specific network).
- All input genes (the union of input CVGs and RVGs).
- Shared input genes (the intersection of input CVGs and RVGs).

We then manually assigned relevant tissues based on each trait's biological domain (e.g., Renal: Kidney). Where relevant tissues could not be clearly defined, traits were excluded from the analysis (e.g., anthropometric traits such as height and BMI). Enrichment analysis was performed using the R package TissueEnrich v1.18.0.

Second, for the subset of traits with CVGs or RVGs expressed in a limited number of tissues, we examined the enrichment for genes expressed in trait-relevant tissues (Fig. 5B). Traits were selected by the difference between the mean number of expressed tissues for CVGs and RVGs ($\delta_{nTissues}$), to give the 15 traits with the lowest $\delta_{nTissues}$ (RVGs expressed in fewer tissues) and the 15 traits with the highest $\delta_{nTissues}$ (CVGs expressed in fewer tissues). These traits were filtered to those that could be assigned a trait-relevant tissue based on the biological domain (as above), giving 23 traits. TSEA was performed for the input gene set (CVGs or RVGs) with expression in the fewest number of tissues on average. Enrichment analysis was performed using the R package TissueEnrich v1.18.0.

Finally, TSEA and individual gene expression distributions for CVGs and RVGs associated with glucose metabolism were sourced from the TissueEnrich browser (https://tissueenrich.gdcb.iastate.edu/) using HPA gene expression data, gene symbols for the trait-associated genes, and a background of PCNet 2.0 genes.

### Similarity of trait-specific networks

The similarity of pairs of trait-specific networks ($G_{trait1}:G_{trait2}$) was measured using cosine similarity. First, we identified the union of genes in both networks. Then we took the cosine similarity of common-rare network proximity scores ($NPS_{CR}$) for genes in the union.

### Construction of interactive hierarchical systems maps for neuropsychiatric traits

To identify the biological systems underlying the convergence of CVGs and RVGs in neuropsychiatric disorders, we filtered the trait-specific networks for Alzheimer's disease (AD), bipolar disorder (BD), and autism spectrum disorder (ASD) to include only genes expressed in the brain. Genes were maintained if they had TPM > 1 in at least one of the 13 brain regions of the Human Protein Atlas brain region data (Sjöstedt et al, 2020; Data ref: Human Protein Atlas, 2024).

Systems maps were then constructed using the hierarchical community detection framework HiDeF (Zheng et al, 2021) via the Cytoscape package CyCommunityDetection v1.12.1 (Singhal et al, 2020). We set the maximum resolution to 10, with all other parameters set to their default values. All network visualizations were constructed in Cytoscape v3.10.3 (Shannon et al, 2003) and uploaded to the Network Data Exchange (Pillich et al, 2021) (NDEx) as part of a network set (Common and rare genetic variants show network convergence for a majority of human traits). For interactive exploration of the systems maps and their underlying protein networks, we formatted the networks to comply with the hierarchical network schema for CX2 (HCX, https://cytoscape.org/cx/cx2/hcx-specification), using the NDEx2 Python Client v3.9.0 (https://github.com/ndexbio/ndex2-client), using the following procedure:

1. Import the hierarchy network and the parent trait-specific network in CX2 format.
2. Link genes in each community to the corresponding genes in the parent network.
3. Assign the uuid identifier of the parent network as an attribute of the HCX hierarchy network.
4. Upload the HCX network to NDEx.
5. Log in to Cytoscape Web (Ono et al, 2025) (web.cytoscape.org) to browse the network. Specific communities can be selected, and the corresponding protein subnetwork, here from PCNet 2.0, will be dynamically displayed.

### Functional enrichment and annotation of trait-specific networks and hierarchical systems maps

Functional enrichments for the glucose measurement trait-specific network were calculated using the g:Profiler (Kolberg et al, 2023) website (https://biit.cs.ut.ee/gprofiler/gost) for KEGG Pathways and Human Phenotypes. Analysis was performed using human NCBI Gene IDs, the set of annotated genes as the background, and the BH correction for multiple hypothesis testing. Only enrichments with an intersection size >1 were considered.

For the AD, BD, and ASD systems maps, all communities were annotated using g:Profiler via the Cytoscape package CyCommunityDetection v1.12.1 (Singhal et al, 2020). Enrichments were considered across a range of functional databases, including Gene Ontology, Reactome, CORUM, and WikiPathways. We set the maximum gene set size to 2000, the maximum $p$-value to $1 \times 10^{-4}$, and the minimum overlap fraction to 0.05. For visualization purposes, we retained annotated communities with at least five genes. Communities without a significant enrichment were labeled NA.

## Data availability

All common and rare variant associations were sourced from the GWAS Catalog (https://www.ebi.ac.uk/gwas/) and RAVAR (http://www.ravar.bio). Associated study accession numbers are listed in Dataset EV1. Computer code used for data processing, analysis, and visualization: GitHub (https://github.com/sarah-n-wright/CARVA) and Zenodo (https://doi.org/10.5281/zenodo.18511481). PCNet 2.0 network: NDEx (https://doi.org/10.18119/N9JP5J). PCNet 2.2 network: NDEx (https://doi.org/10.18119/N9960N). STRING & HumanNet networks: NDEx (https://doi.org/10.18119/N95C9J). Trait-specific networks generated: NDEx (Common and rare genetic variants show network convergence for a majority of human traits).

The source data of this paper are collected in the following database record: biostudies:S-SCDT-10_1038-S44319-026-00733-4.

## Peer review information

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

## Acknowledgements

This work was supported by the following grants from the National Institutes of Health: NIMH U01 MH115747 to TI, and NIDA P50 DA037844 to TI. This work was also supported by the following grants from the California Institute for Regenerative Medicine: ReMIND DISC4-16322 and ReMIND DISC4-16377.

## Author contributions

**Sarah N Wright**: Conceptualization; Data curation; Software; Formal analysis; Validation; Visualization; Methodology; Writing—original draft; Writing—review and editing. **Jane Yang**: Methodology; Writing—original draft; Writing—review and editing. **Trey Ideker**: Conceptualization; Resources; Supervision; Funding acquisition; Writing—original draft; Project administration; Writing—review and editing.

Source data underlying figure panels in this paper may have individual authorship assigned. Where available, figure panel/source data authorship is listed in the following database record: biostudies:S-SCDT-10_1038-S44319-026-00733-4.

## Disclosure and competing interests statement

TI is a co-founder, member of the advisory board, and has an equity interest in Data4Cure and Serinus Biosciences. TI is a consultant for and has an equity interest in Ideaya Biosciences and Eikon Therapeutics. The terms of these

arrangements have been reviewed and approved by the University of California, San Diego, in accordance with its conflict-of-interest policies.

# Expanded View Figures

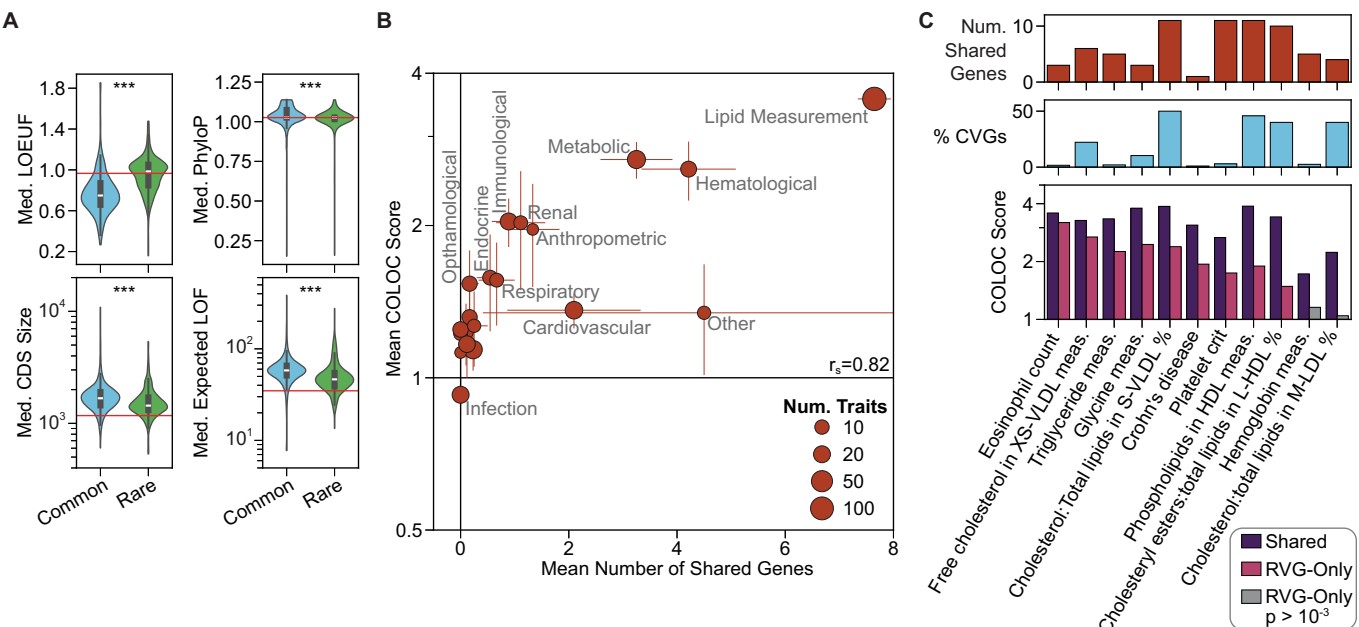

**Figure EV1. Further exploration of gene properties, disease groups, and synthetic associations.**

(A) Comparison of CVG and RVG properties across 373 traits using alternative metrics of gene constraint and gene size. The value for each set of CVGs or RVGs was taken as the median of all genes in each set (Wilcoxon Signed-Rank test, BH correction). The center box plots show the median property value and interquartile range (IQR), with the lower and upper whiskers extending to $Q1 - 1.5IQR$ and $Q3 + 1.5IQR$. The violins extend to the minimum and maximum observations. Red lines indicate the median value for all protein-coding genes. ***$q < 1 \times 10^{-5}$. $q_{MedLOEUF} = 5.2 \times 10^{-34}$, $q_{MedPhyloP} = 4.2 \times 10^{-11}$, $q_{MedCDSSize} = 8.4 \times 10^{-9}$, $q_{MedExpectedLOF} = 1.4 \times 10^{-13}$. (B) Comparison of the COLOC score and the number of shared CVG and RVG genes for traits within each biological domain. Points represent the mean values within each domain, with error bars representing the standard errors. Point size indicates the number of traits per domain. (C) Estimation of the impact of synthetic associations on COLOC scores. Ten random traits with at least 3 shared genes identified by CV and RV association studies, and Crohn's disease (a trait with a known synthetic association), were analyzed by removing shared genes from the set of CVGs. The top bar plot shows the number of shared genes for each trait, and the middle bar plot shows the shared genes as a percentage of all CVGs. The bottom bar plot shows the COLOC scores for each trait, allowing shared genes or treating shared genes as RVGs only. Colored bars represent significant network colocalizations ($p < 10^{-3}$).

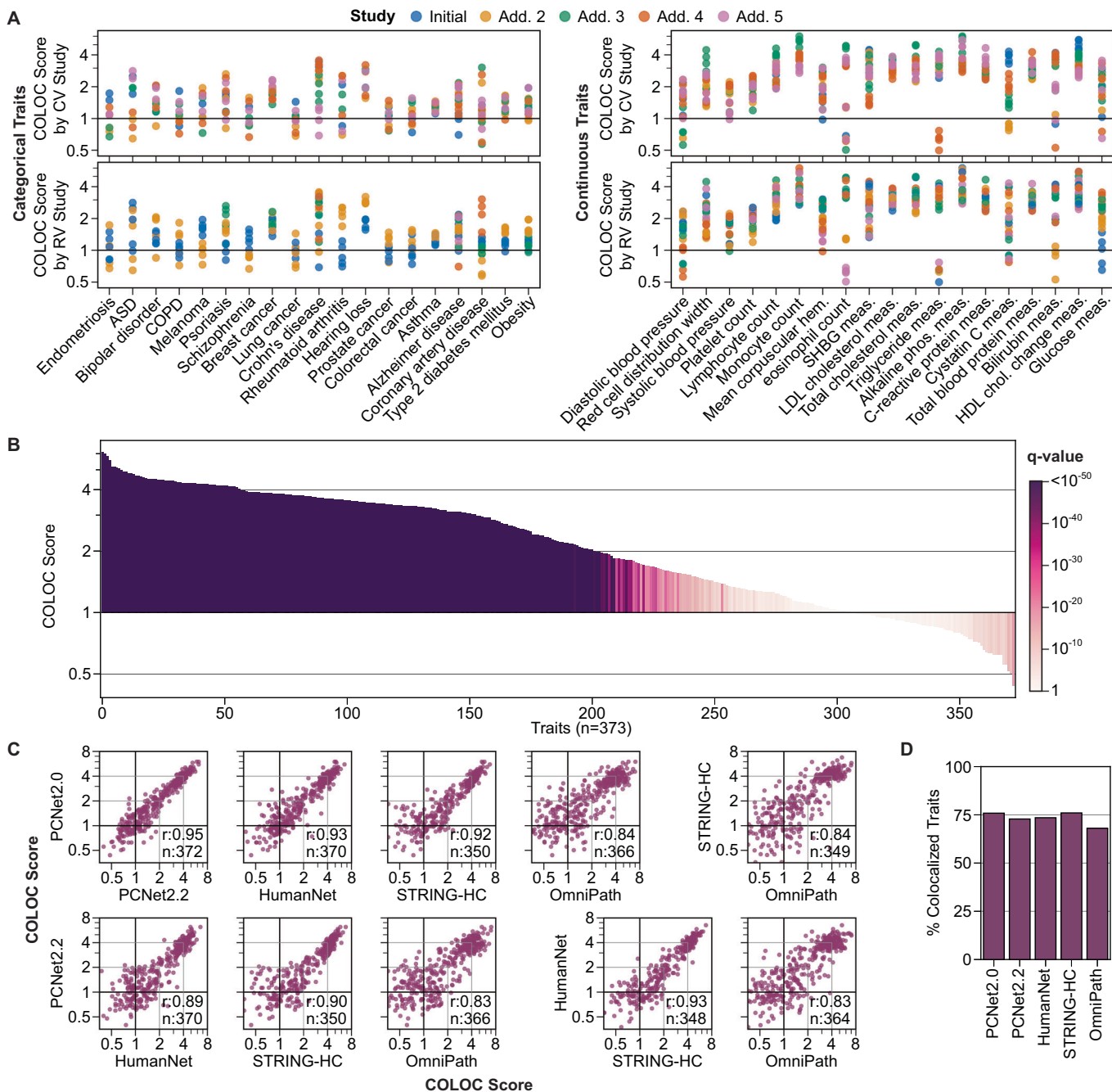

**Figure EV2. Robustness of network convergence across different studies and networks.**

(A) COLOC scores for categorical traits with ≥10 rare-common study combinations (left, $n = 19$) and continuous traits with ≥20 common-rare study combinations (right, $n = 19$). The top plots are colored by CV study, and the bottom plots are colored by RV study. ASD: autism spectrum disorder, COPD: Chronic Obstructive Pulmonary Disease. Point colors indicate whether each study was part of the initial or an additional study combination. (B) Network colocalization results for 373 traits after taking the highest COLOC score from all available common-rare study analyses per trait. Traits are ordered by COLOC score, and bar color indicates q-value (BH correction) of the COLOC score based on a Z-test against 1000 permutations. (C) Pairwise comparison of COLOC scores across five biological networks. Correlation (r) calculated using Spearman correlation. The number of traits that could be assessed by both networks is given by n. (D) For each network, the fraction of evaluated traits with a nominally significant COLOC score ($p < 0.05$). Related to Fig. 3.

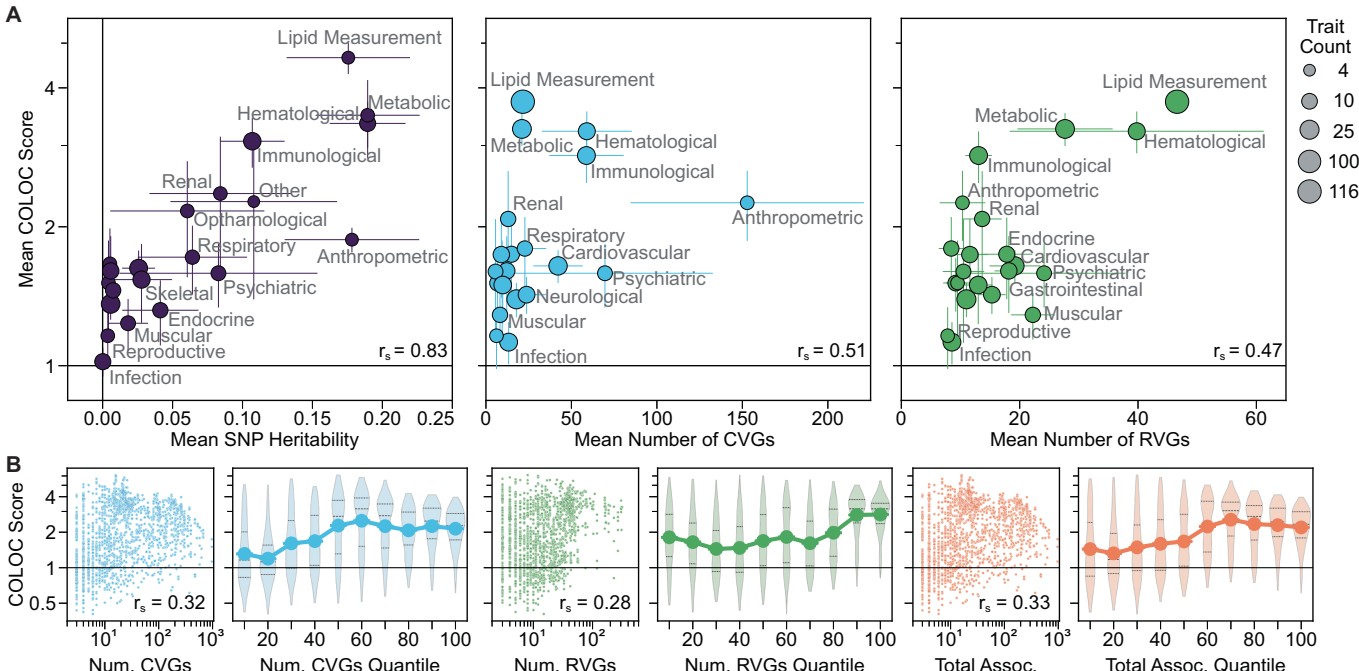

**Figure EV3. Network convergence as a function of gene set and biological properties.**

(A) Correlation of mean COLOC score across biological domains with trait SNP heritability, number of CVGs, and number of RVGs. Points represent the mean values within each domain, with error bars representing the standard errors. Point size indicates the number of traits per domain. Heritability results are shown for the subset of 197 traits for which a SNP heritability estimate was available, and the number of gene results are shown for the 373 best trait study pairs (Fig. EV2B). (B) COLOC score as a function of the number of underlying CVG and RVG associations for all combinations of initial and additional studies ($n = 1634$). Total association count is the total number of distinct CVGs and RVGs per trait. Spearman correlation reported. Line plots show the mean COLOC score for each quantile, and violin plots show the distribution of COLOC scores within each feature quantile, with the median and quartiles represented by horizontal lines. Violins extend to the minimum and maximum observed values.

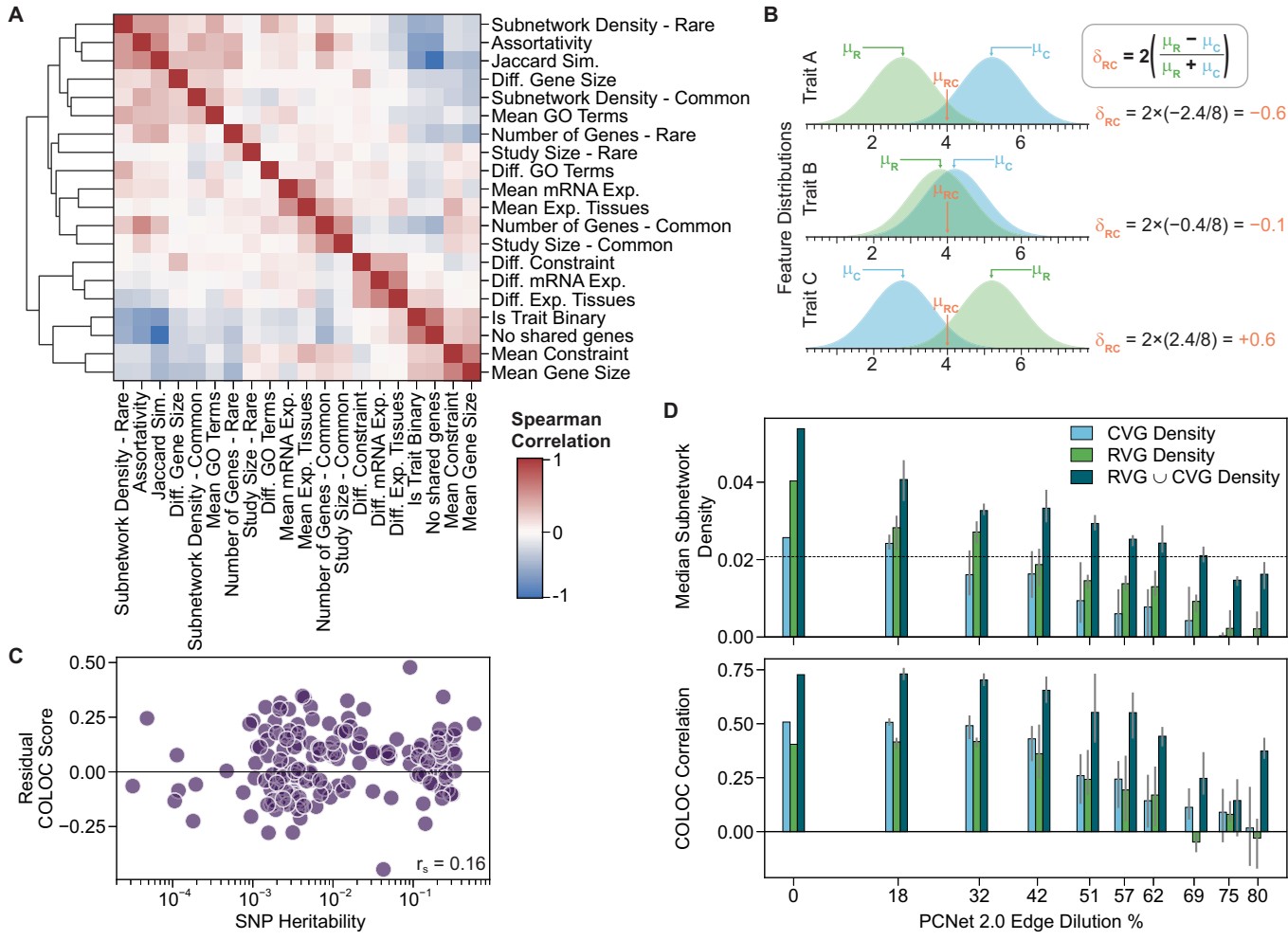

**Figure EV4. Expanded analysis of gene and trait features.**

(A) Correlation of gene and trait features over 1634 common-rare study combinations for 373 traits. Jaccard similarity was calculated between the sets of CVGs and RVGs. Gene features were summarized by the overall mean of CVGs and RVGs, and the difference between CVGs and RVGs. Constraint: gene selective constraint defined by $s_{het}$ estimates. Exp. Tissues: Number of tissues with mRNA expression >1 TPM. (B) Schematic showing the calculation of mean ($\mu$) and difference ($\delta$) metrics for gene features. Distributions represent the hypothetical feature values for all CVGs (blue) and RVGs (green) of a given trait. Where the RVGs have higher average property values, the $\delta$ value will be positive. (C) Residual COLOC score as a function of estimated SNP heritability for the 197 traits with available heritability data. Residual COLOC score calculated as the difference between the observed COLOC score and the COLOC score predicted by the Elastic Net regression model following five-fold cross-validation. Spearman correlation reported. (D) Subnetwork density of RVGs and CVGs, and Spearman correlation with COLOC score for partitioned GWAS test traits across progressively randomized versions of PCNet 2.0. Three series of randomized networks were generated by progressively diluting PCNet 2.0 via degree-preserving edge swaps, with subnetwork density calculated with each network structure. Dilution percent is determined as 100 × (1 − J), where J is the Jaccard similarity of edges in the original and randomized networks. Bar plots show the values for the original network (dilution = 0%, $n = 1$) or the median value for the diluted networks (dilution > 0%, $n = 3$), with error bars indicating the minimum and maximum values. Related to Fig. 4.

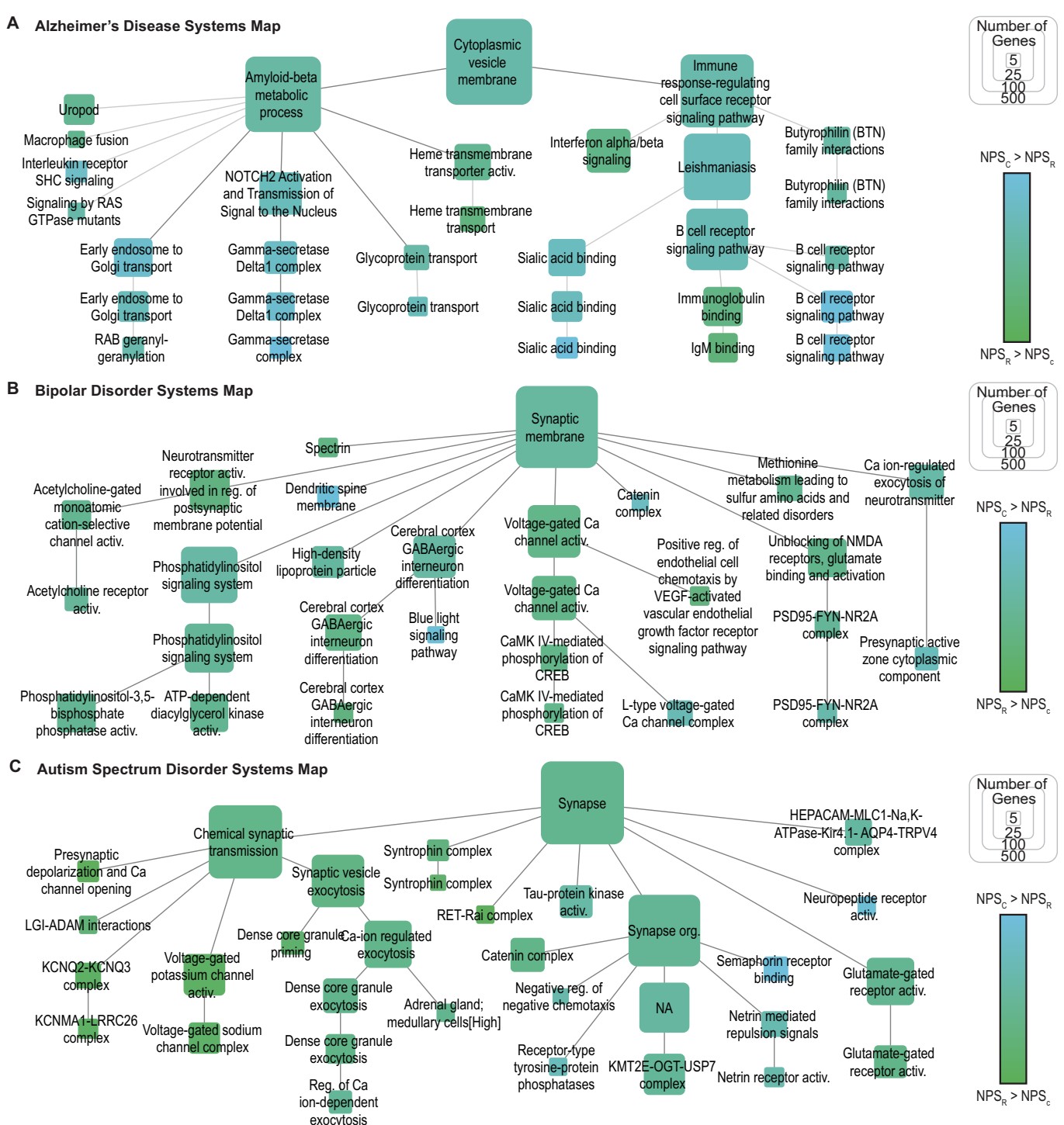

**Figure EV5. Systems maps constructed from the trait-specific networks for select neuropsychiatric traits.**

Maps generated via hierarchical community detection of the trait-specific networks for (**A**) Alzheimer's disease, (**B**) bipolar disorder, and (**C**) autism spectrum disorder, and annotated using g:Profiler (Methods). Node size indicates the number of genes per system, and node color represents the ratio of NPS$_C$ and NPS$_R$ for genes within the system. Systems with more than five genes and that could be annotated using g:Profiler are displayed. Underlying trait-specific networks were filtered to genes expressed in at least one brain region with TPM > 1 (HPA). Related to Fig. 6.

