## [Peer Review File · EMBO Reports]

Common and rare genetic variants show network convergence for a majority of human traits

Sarah Wright, Jane Yang, and Trey Ideker

Corresponding author(s): Trey Ideker (tideker@ucsd.edu)

Review Timeline:

Transfer Date:	18th Nov 25
Editorial Decision:	25th Dec 25
Revision Received:	25th Jan 26
Accepted:	12th Feb 26

Editor: Yehu Moran

**Transaction Report: This manuscript was transferred to
EMBO reports following peer review at Review Commons.**

Review
COMMONS

Review #1

1. Evidence, reproducibility and clarity:

Evidence, reproducibility and clarity (Required)

In this paper, Wright, Yang, and Ideker analyze whether rare variant burden tests and GWAS converge on similar genes from the perspective of gene networks. In particular, they consider a large number of association studies from the GWAS catalog and RAVAR and use these to compute per-trait, per-study-design vectors of weights for each gene based on propagating (-log₁₀ transformed) p-values through a gene network. With these vectors in hand, they then ask whether the values based on GWAS and burden tests are more similar than expected by chance and then perform a number of analyses about what properties impact the amount of overlap and the network scores themselves. Overall, I found the paper to be interesting and well-written. It is a nice contribution to the literature. I do have a few important technical concerns that I list below, and I also feel that the authors could have done more in terms of presenting interpretations of their results. I also list a number of more minor comments below in the hopes that they're useful to the authors.

****Major comments:****

- I am concerned about the LOEUF results presented in the paper and any analyses that used LOEUF or had interpretations based on LOEUF (e.g., Figure 1C). In particular, in the main text it's stated that G6PC2 and KCNK16 both have negative LOEUF scores, and hence must be strongly constrained. LOEUF scores are always strictly non-negative (they're defined as the upper limit of a confidence interval on a ratio of strictly non-negative numbers). It's possible that what is getting reported in the main text is the Normal-transformed version (as described in the supplement) but it would be good to be clear about that as people have a sense of what different values of LOEUF correspond to. And even if this is the case, there must be a sign flip somewhere because G6PC2 has a LOEUF of 1.679

(https://gnomad.broadinstitute.org/gene/ENSG00000152254?dataset=gnomad_r4) and KCNK16 has a LOEUF of 1.463

(https://gnomad.broadinstitute.org/gene/ENSG00000095981?dataset=gnomad_r4) and so neither has any evidence whatsoever of constraint. In contrast, SLC30A8 is stated to be less constrained than G6PC2 or KCNK16, but it has a lower LOEUF of 0.921

(https://gnomad.broadinstitute.org/gene/ENSG00000164756?dataset=gnomad_r4), so if anything there is evidence of it being more constrained (albeit only weakly constrained).

This is a much more minor point, but in addition to all of these technical concerns, LOEUF is a difficult to interpret metric of constraint as it confounds signal and power (see <https://doi.org/10.1038/s41588-024-01820-9>) and I would recommend instead using s_{het} (<https://doi.org/10.5281/zenodo.10403680>).

- I also have a technical concern about deriving the expected overlap for the COLOC score. In particular, permuting the NPS_C values breaks any correlation in the NPS_C values themselves induced by the graph structure (e.g., presumably all of the nodes near a strong GWAS hit have fairly high NPS_C values, and so high NPS_C values should cluster in the network, whereas permuting should evenly spread them across the network). It seems more intuitive to me to instead permute the gene-level p-values themselves and then recompute the NPS_C values for each permutation.

- This is a stylistic comment, and so the authors should feel free to ignore it if it's not useful, but I felt that many parts of the paper could be strengthened by including some higher level interpretation of the results (even if speculative, as long as appropriately stated as such). For example, many of the results suggest that differences in GWAS power drive some of the differences in network convergence (as hinted at by differences in continuous vs. categorical traits, and differences due to differences in heritability). Another example -- what does it mean that different studies produce different degrees of COLOC for some traits? Is this purely random? It is due to different cohort compositions and some kind of GxE or GxG interaction?

****Minor comments:****

- At the top of p. 4 it would be good to add a caveat about selection bias. In particular, these genes are those for which there's statistical evidence of an association, and so any technical factors that contribute to power would appear to be correlated with discovered RVGs or CVGs, but these technical factors might not correlate with the actual set of genes with trait-affecting rare or common variants. For example, longer genes will have more "shots on goal" in GWAS and so are more likely to be discovered, even if they are not actually more important in general. It would also be interesting to repeat the gene length analysis but using the CDS length (or gnomAD's expected number of LoFs) as that is more relevant to burdens test power.

- This is another stylistic comment, so the authors can feel free to ignore, but I found the nomenclature of "trait network" to be a bit confusing. It's not a network on traits (e.g.,

nodes are traits), and instead it's more about the overlap between genes that have network evidence from both CVGs and RVGs.

- Figure 4A suggests that maybe overlap is being driven (at least in part) by power. Does a similar trend hold when using the number of genes discovered by GWAS instead of heritability? This would be consistent with <https://doi.org/10.1101/2024.12.12.628073>, where it's suggested that GWAS discovers an essentially random subset of the set of genes that contribute to a trait, and so increasing the number of genes discovered by GWAS would increase the likelihood/degree of overlap.

- It would be good to include a caveat about the significance measure in Table 1. In particular, I suspect that many of these predictors are highly correlated (and hence nearly collinear), and so the p-values / confidence intervals reflect not only the uncertainty in the marginal association, but the extent to which any association can be explained away by other predictors. E.g., "Number of genes" might have a large effect, but if it correlates with "Categorical trait", then many elastic net models will drop one predictor or the other.

- It would be good to include a discussion of <https://doi.org/10.1101/2025.02.24.639925> which also investigates the convergence of rare and common variant.

2. Significance:

Significance (Required)

This is an important paper and provides new insights onto a problem that has been of recent interest to the statistical genetics community. The network approach to the problem is novel and a strength of the paper. The paper could be improved by better connecting its results to similar papers in the field (e.g., <https://doi.org/10.1186/s13073-023-01253-9>, <https://doi.org/10.1038/s41586-022-05684-z>, <https://doi.org/10.1101/2024.12.12.628073>, and <https://doi.org/10.1101/2025.02.24.639925>), and there are a few (easily fixable) potential technical flaws with the work. The paper should be interesting to a broad segment of the statistical genetics / human genetics community, and should be useful in the interpretation and design of association studies.

For context, my expertise is in statistical genetics, population genetics, polygenic scores, and statistics.

3. How much time do you estimate the authors will need to complete the suggested revisions:

Estimated time to Complete Revisions (Required)

(Decision Recommendation)

Between 1 and 3 months

4. Review Commons values the work of reviewers and encourages them to get credit for their work. Select 'Yes' below to register your reviewing activity at Web of Science Reviewer Recognition Service (formerly Publons); note that the content of your review will not be visible on Web of Science.

Yes

Review #2

1. Evidence, reproducibility and clarity:

Evidence, reproducibility and clarity (Required)

The manuscript of Wright et al describes a very important phenomenon, how common and rare variants are connected in molecular networks across multiple disease traits. The manuscript text is clear with a good flow. The figures present complex analyses in a visually understandable way with relevant and helpful captions. The authors executed a state-of-the-art network biology analysis with a well-documented, community standard and novel workflow. The output data (networks) was also shared as expected based on best practices. In each analytical step, adequate statistical analyses describe key features and properties of their finding. The Discussion is surprisingly short compared to the complexity and length of the Result section. Limitations of the study are embedded in generally positive short paragraphs. Comparison to similar approaches are missing.

Here, I would like to raise a few points to increase the validity of the findings and improve the biological insight we can gain from this work:

- The network analysis work is based on the main author's previous integrated and combined network resource, PCNet2. While this is an excellent resource for protein-protein interactions, some key analysis should be also done as a control with other network sources (for example using only one key component of PCNet2 (eg STRING) and

one similar large network not part of PCNet 2 (eg OmniPath).

- It should be clarified throughout that this work focuses on coding region variants (SNPs), and, as explained in the Discussion very briefly, left out non-coding SNPs - which are the majority of common and rare variants. Because this understandable technical filter, the conclusions and main messages should be accordingly toned down.
- It would be interesting to add some thoughts to the Discussion how this analysis could be done with gene regulatory networks, and whether the authors expect that the identified phenomena is limited to protein-protein interaction networks, or will probably true for regulatory networks as well.
- The authors analysed tissue specificity and Human Protein Atlas based tissue-specific enrichment, but given the currently already available single-cell datasets from some of the published Human Cell Atlas projects, a more fine-grained analysis would be helpful. Such large cell-type specific expression data overlaid with the networks would be important to dissect and check whether the observed network colocalizations are not the product of aggregated (tissue-level) interactions.
- At multiple analyses, interesting differences between disease groups (eg infection, cancer, neurodegeneration) are pointed out. However, these differences are not explored or explained later. This is a missed opportunity to demonstrate the applicability of the authors' work to gain new biological insight on the systems-level features of various diseases. Some comparative results with explanation, and a new paragraph in the Discussion could cover this.

****Referees cross-commenting****

I agree with the points raised by the other reviewers. Also, good to see that all the three of us felt it is a good study that needs a bit of revision but not substantial.

2. Significance:

Significance (Required)

Overall, this is a great manuscript and well conducted study. The core concept is very important for any researcher working on systems genomics, network biology and higher-level diseases systems. As the work follows best practices in systems biology, the paper will be very interesting for researchers at various career stages and for teaching/training

purposes as a good example study.

Tamas Korcsmaros

3. How much time do you estimate the authors will need to complete the suggested revisions:

Estimated time to Complete Revisions (Required)

(Decision Recommendation)

Between 3 and 6 months

No

Review #3

1. Evidence, reproducibility and clarity:

Evidence, reproducibility and clarity (Required)

The genetic architecture of complex human traits and diseases remains an open challenge. GWAS have been extensively used over the past decade to discover associations between common variants (typically MAF >1%) and human phenotypes, while whole-exome sequencing (WES) studies have been employed to identify such associations for rare variants. Several pioneering studies have demonstrated, for selected diseases, that associated common and rare variants can converge on shared biological processes. This study by Wright et al. is the first to systematically investigate this intriguing question. To address it, the authors applied a cutting-edge systems biology approach based on network colocalization. Analyzing 373 traits, they found that 254 showed significant network convergence between genes implicated by GWAS ("common variant genes" - CVGs) and by WES ("rare variant genes" - RVGs). This result is especially remarkable given that, for approximately 50% of the traits (189/373), there was no direct gene overlap between the

CVG and RVG sets.

The study deals with an important and unexplored problem. Overall, it is very well written, and the results are presented very well. A few issues must be addressed before it can be accepted.

****Major Comments:****

- Shared genes between the CVG and RVG sets:

One key issue should be addressed regarding the analysis relating to the genes shared between the CVG and RVG sets. It is known that in some cases, common SNPs identified in GWAS may tag nearby causal rare variants ('synthetic associations'). In such cases, the shared genes may, in fact, be influenced solely by the rare variants. This is particularly important given the strong correlation observed between the number of shared genes for the trait and its COLOC score for the CVG and RVG networks. The authors should rule out this possibility.

- Sensitivity to the network structure

1) Previous works have shown that similar tasks (e.g., active module identification) highly depends on the structure of the network. To show the contribution of the network structure, please provide an analysis of the impact of the selected network on the main results. For example, gradually drop a larger fraction of random edges and see how it affects the results. Alternatively, gradually apply higher number of edges swaps to introduce higher noise into the network. More ideas were applied in PMID: 33782690

2) In the linear regression estimates, the subnetwork density seems to be important in the prediction, and that density is also highly affected by the network structure. Systematic evaluation of different networks is relevant for this part as well.

- P-value justification.

The significance of the COLOC scores that underlie the study are based on computing COLOC score for 1000 randomized networks and then converting the Z-score a p-value using a assuming the distribution is standard normal. Please provide evidence that this assumption holds.

- Figure 2D: The dots are colored by significance (q-value). If so, how can some highly significant traits (dark purple) exhibit a COLOC score close to 1? Please clarify.

****Other comments:****

- **Gene prioritization:**

"Compared to the original input genes, these prioritized trait genes and the full trait networks showed better enrichment for genes expressed in trait-relevant tissues, demonstrating that the network approach prioritizes functionally relevant genes (Figure 3F)": The term "prioritized" is confusing here since it was not defined. Please move the definition of the term from the legend to the main text where the term appears in the first time.

- **Figure 5D:**

1) The enrichment analysis for the glucose measurement trait network uses the entire set of human genes as the background. It would be valuable to show whether the identified enriched processes are specific to the delineated network (e.g., whether trait-associated but non-prioritized genes show weaker enrichments).

2) What was the size of the glucose measurement trait network? From the fact that a threshold of >20 gene in the intersection was used it must be very large.

- **Figure 2c:** Only a few categorical traits exhibit very high COLOC scores. What are the top 10 categorical traits? Do they share any common features that might explain their high scores?

- **Page 10:** Two genes are reported with negative LOEUF scores. However, by definition, LOEUF scores are positive. Please clarify-were these scores processed or filtered incorrectly?

- **Table S2:** There are entries where there is no overlap ($n_{\text{Shared}} = 0$) between CVGs and RVGs, yet the p-value for overlap (p_{Shared}) is significant. Please clarify.

2. Significance:

Significance (Required)

The study deals with an important and unexplored problem.

We found the analysis methodology novel and very interesting, more than the biological results.

3. How much time do you estimate the authors will need to complete the suggested revisions:

Estimated time to Complete Revisions (Required)

(Decision Recommendation)

Between 1 and 3 months

No

Full Revision

Manuscript number: RC-2025-03100

Corresponding author(s): Trey Ideker

1. Point-by-point description of the revisions

This section is mandatory. Please insert a point-by-point reply describing the revisions that were already carried out and included in the transferred manuscript.

Reviewer #1 (Evidence, reproducibility and clarity (Required)):

In this paper, Wright, Yang, and Ideker analyze whether rare variant burden tests and GWAS converge on similar genes from the perspective of gene networks. In particular, they consider a large number of association studies from the GWAS catalog and RAVAR and use these to compute per-trait, per-study-design vectors of weights for each gene based on propagating (-log₁₀ transformed) p-values through a gene network. With these vectors in hand, they then ask whether the values based on GWAS and burden tests are more similar than expected by chance and then perform a number of analyses about what properties impact the amount of overlap and the network scores themselves. Overall, I found the paper to be interesting and well-written. It is a nice contribution to the literature. I do have a few important technical concerns that I list below, and I also feel that the authors could have done more in terms of presenting interpretations of their results. I also list a number of more minor comments below in the hopes that they're useful to the authors.

Major comments:

- I am concerned about the LOEUF results presented in the paper and any analyses that used LOEUF or had interpretations based on LOEUF (e.g., Figure 1C). In particular, in the main text it's stated that G6PC2 and KCNK16 both have negative LOEUF scores, and hence must be strongly constrained. LOEUF scores are always strictly non-negative (they're defined as the upper limit of a confidence interval on a ratio of strictly non-negative numbers). It's possible that what is getting reported in the main text is the Normal-transformed version (as described in the supplement) but it would be good to be clear about that as people have a sense of what different values of LOEUF correspond to. And even if this is the case, there must be a sign flip somewhere because G6PC2 has a LOEUF of 1.679 (https://gnomad.broadinstitute.org/gene/ENSG00000152254?dataset=gnomad_r4) and KCNK16 has a LOEUF of 1.463 (https://gnomad.broadinstitute.org/gene/ENSG00000095981?dataset=gnomad_r4) and so neither has any evidence whatsoever of constraint. In contrast, SLC30A8 is stated to be less constrained than G6PC2 or KCNK16, but it has a lower LOEUF of 0.921 (https://gnomad.broadinstitute.org/gene/ENSG00000164756?dataset=gnomad_r4), so if anything there is evidence of it being more constrained (albeit only weakly constrained). This is a much more minor point, but in addition to all of these technical concerns, LOEUF is a difficult to interpret metric of constraint as it confounds signal and power

(see <https://doi.org/10.1038/s41588-024-01820-9>) and I would recommend instead using s_{het} (<https://doi.org/10.5281/zenodo.10403680>).

The reviewer is correct that the LOEUF values reported are incorrect. LoF Z-scores were mistakenly reported in place of LOEUF. We thank the reviewer for flagging this issue. We have updated this analysis to utilize s_{het} , as suggested, which is now included in the **updated Figure 1c**. We include analysis of the corrected LOEUF as well as evolutionary conservation (PhyloP) in the **new Supplemental Figure S1a**. The results section *Identification of 373 human traits with common and rare variant associations* now reads:

“Rare variant study populations tended to be larger, with a median of 210,000 individuals, compared to a median of 136,000 individuals for common variant studies (**Figure 1c**, $q = 6.8 \times 10^{-8}$). CVGs were more mutationally constrained ($q = 3.5 \times 10^{-29}$), as evidenced by higher values of s_{het} , a continuous metric that measures the reduction in fitness for heterozygous carriers of LOF mutations in a given gene [16]. In addition, CVGs had longer gene regions ($q = 1.6 \times 10^{-50}$) and more functional annotations ($q = 2.2 \times 10^{-5}$), with expression in a greater number of tissues ($q = 4.9 \times 10^{-4}$). The differences between CVGs and RVGs were consistent when examining alternative metrics of gene length and mutational constraint (**Figure S1a**). For example, CVGs had longer CDS sequences ($q = 8.4 \times 10^{-9}$) and showed stronger evolutionary conservation (phyloP, $q = 4.2 \times 10^{-11}$). These distinct gene properties likely reflect a combination of differences in variant impacts, as well as selection biases from the sequencing and statistical association methodologies.”

In addition, we have updated all our regression analysis and associated figures (**updated Figure 4, Supplemental Figure S4**) to use the s_{het} metric rather than LOEUF, and removed incorrect results from the results section *Tissue-specific RVGs are associated with stronger network convergence*. Details for the processing of the new metrics have been added to the Methods section *Data Curation*:

“Gene mutational constraint estimates based on loss-of-function (LoF) mutations were sourced from gnomAD v4.1 [54] (Constraint metrics TSV, <https://gnomad.broadinstitute.org/data>). For each gene, we extracted the LOEUF (loss-of-function observed/expected upper bound fraction) [54,55], the synonymous intolerance Z-score [55,56], and the expected number of LoF mutations. The LOEUF metric provides a continuous metric of a gene’s intolerance to loss-of-function variants, with a threshold of LOEUF < 0.6 recommended for defining LoF-constrained genes. For each distinct gene symbol, we prioritized entries associated with canonical and MANE Select transcripts, followed by prioritization of entries with an NCBI Gene ID assigned.

The lengths of coding sequences for each gene (CDS) were derived from GENCODE [57] v46 Basic Gene Annotation. Positional gene conservation scores (phyloP) were sourced from the UCSC Genome Browser [58], calculated via multiple alignment of 29 vertebrate species to the hg38 human genome. PhyloP scores from the reference chromosomes were aggregated for each transcript using BEDOPS v2.4.41 [59]. The final gene conservation score was defined per gene as the mean phyloP across all positions within the gene’s CDS.”

Figure 1. Curation and analysis of common and rare variant associations for 373 traits. **c** Comparison of CVG and RVG properties across 373 traits (Wilcoxon Signed-Rank Test, BH correction). For gene-level properties, the value for each set of CVGs or RVGs was taken as the median of all genes in each set. The center box plots show the median property value and interquartile range (IQR), with the lower and upper whiskers extending to $Q1 - 1.5IQR$ and $Q3 + 1.5IQR$. The violins extend to the minimum and maximum observations. Red lines indicate the median value for all protein-coding genes, where applicable. *** $q < 1 \times 10^{-5}$, ** $q < 1 \times 10^{-3}$, n.s. $q > 0.05$.

Figure S1. Further exploration of gene properties, disease groups, and synthetic associations. **a** Comparison of CVG and RVG properties across 373 traits using alternative metrics of gene constraint and gene size. The value for each set of CVGs or RVGs was taken as the median of all genes in each set (Wilcoxon Signed-Rank Test, BH correction). The center box plots show the median property value and interquartile range (IQR), with the lower and upper whiskers extending to $Q1 - 1.5IQR$ and $Q3 + 1.5IQR$. The violins extend to the minimum and maximum observations. Red lines indicate the median value for all protein-coding genes. *** $q < 1 \times 10^{-5}$.

- I also have a technical concern about deriving the expected overlap for the COLOC score. In particular, permuting the NPS_C values breaks any correlation in the NPS_C values themselves induced by the graph structure (e.g., presumably all of the nodes near a strong GWAS hit have fairly high NPS_C values, and so high NPS_C values should cluster in the network, whereas permuting should evenly spread them across the network). It seems more intuitive to me to instead permute the gene-level p-values themselves and then recompute the NPS_C values for each permutation.

The reviewer is correct that the permutation approach does not maintain the correlation between nearby nodes in the network, which could skew the results. For example, if the permutation process systematically underestimated the expected size of the colocalized trait network, then the COLOC scores would be inflated even for random inputs. We have now expanded on previous testing (PMID: 36653526, Fig. 5) to confirm that we do not see inflation of COLOC scores with random genes. Using both GO-derived (**new Supplemental Figure S8a**) and GWAS-derived gene sets (**new Supplemental Figure S8b**), we observe a consistent decrease in COLOC score as the genes are replaced with degree-matched random genes. When all gene sets are completely randomized (dilution = 100%), the COLOC scores do not significantly differ from the expected COLOC value of 1. We reference these figures in the *Methods* section *Optimization and Benchmarking of Q-NetColoc* as:

“Input gene set dilution testing. To test the response of NetColoc to the introduction of random error in the gene sets, we progressively diluted the input data with degree-matched random genes. All dilution analyses were performed using the parameters $\beta = 20$, $\tau = 1$, and $\tau^* = 3$. For testing B-NetColoc, we utilized gene sets derived from the Gene Ontology (GO), with human GO gene associations sourced from NCBI using goatools v1.5.2 [53] on July 11, 2024. All genes associated with a specific GO term were defined as associated with all parent GO terms. GO terms with between 200 and 250 associated genes were selected from each ontology branch to give 48 biological process (BP) gene sets, 22 cellular compartment (CC) gene sets, and 16 molecular function (MF) gene sets. All GO gene sets were subsampled to 150 genes, partitioned into two non-overlapping sets, and assessed via B-NetColoc. Across the mean of three repeats, all GO gene sets showed a significant COLOC score and were then diluted to contain 25%, 50%, 75% and 100% random genes, with three repeats.

For testing Q-NetColoc, we used the partitioned GWAS test gene sets defined by S2G mapping for Q-NetColoc optimizations, with a threshold of $\theta = 1 \times 10^{-5}$. All partitioned GWAS gene sets with a significant COLOC score ($n = 38$) were then diluted to contain 25%, 50%, 75% and 100% random genes, with 5 repeats. When replacing a given gene with a degree-matched random gene, the association p-value of that gene was given to the random gene.

We observed that COLOC scores consistently decayed as the input gene sets were diluted (**Figure S8a-b**). At 100% dilution, the mean of the distribution of COLOC scores was not significantly different from the null expectation of COLOC = 1 for GO gene sets ($\mu_{\text{COLOC}} = 0.983$, $p_{\text{GO}} = 0.055$, 1-sample t-test) or partitioned GWAS gene sets ($\mu_{\text{COLOC}} = 1.03$, $p_{\text{GWAS}} = 0.36$, 1-sample t-test), indicating that the permuted null distributions used for assessing network colocalization are well calibrated.”

Figure S8. Benchmarking of Network Colocalization. **a-b** COLOC score as a function of gene set dilution for **(a)** B-NetColoc with GO gene sets and **(b)** Q-NetColoc with partitioned GWAS gene sets. The dilution percent indicates the percent of input gene set genes replaced with degree-matched random genes. Results for GO gene sets averaged over three repeats, and results for partitioned GWAS averaged over five repeats. Box plots show the median COLOC score and interquartile range (IQR), with the lower and upper whiskers extending to $Q1 - 1.5IQR$ and $Q3 + 1.5IQR$. Histograms show the distribution of COLOC scores for completely randomized gene sets; p-values are calculated using a 1-sample t-test with the null hypothesis $\mu_{\text{COLOC}} = 1$.

- This is a stylistic comment, and so the authors should feel free to ignore it if it's not useful, but I felt that many parts of the paper could be strengthened by including some higher level interpretation of the results (even if speculative, as long as appropriately stated as such). For example, many of the results suggest that differences in GWAS power drive some of the differences in network convergence (as hinted at by differences in continuous vs. categorical traits, and differences due to differences in heritability). Another example -- what does it mean that different studies produce different degrees of COLOC for some traits? Is this purely random? It is due to different cohort compositions and some kind of GxE or GxG interaction?

We have significantly reworked our *Discussion* to include additional interpretation and context for key results. For example:

“Overall, the extent of convergence between common and rare variants was consistent across different association studies and network resources (**Figure 3b**, **Figure S2c-d**). However, for some traits, the selection of input CV and RV studies influenced the observed convergence (**Figure 3c**, **Figure S2a**). Such discrepancies likely result from differences in study power, population cohorts, confounding variables, or trait definitions. While we focused on study pairs with the strongest convergence, further analysis of cases where convergence depends on the cohort composition could prove informative for translational applications. Similarly, some traits showed variable results with different networks, possibly due to the varying representation of systems and datatypes within each resource [17]. Network convergence could also be evaluated using NetColoc with targeted network resources, such as regulatory or signaling-specific networks [17,45,46].”

Minor comments:

- At the top of p. 4 it would be good to add a caveat about selection bias. In particular, these genes are those for which there's statistical evidence of an association, and so any technical factors that contribute to power would appear to be correlated with discovered RVGs or CVGs,

but these technical factors might not correlate with the actual set of genes with trait-affecting rare or common variants. For example, longer genes will have more "shots on goal" in GWAS and so are more likely to be discovered, even if they are not actually more important in general. It would also be interesting to repeat the gene length analysis but using the CDS length (or gnomAD's expected number of LoFs) as that is more relevant to burdens test power.

We have added analysis of the CDS length and expected number of LoFs as **new Supplemental Figure S1a**, which we reference in the results section *Identification of 373 human traits with common and rare variant associations*:

"The differences between CVGs and RVGs were consistent when examining alternative metrics of gene length and mutational constraint (**Figure S1a**). For example, CVGs had longer CDS sequences ($q = 8.4 \times 10^{-9}$) and showed stronger evolutionary conservation (phyloP, $q = 4.2 \times 10^{-11}$). These distinct gene properties likely reflect a combination of differences in variant impacts, as well as selection biases from the sequencing and statistical association methodologies."

And in the *Discussion*:

"Several limitations should be considered when interpreting these results. Some distinct properties of CVGs and RVGs may result from underlying selection biases and limitations of the SNP-to-Gene mapping procedure. For example, larger genes have more opportunities to be tagged by a causal variant in GWAS, which is reflected in our finding that CVGs tended to be longer (**Figure 1c**). This bias may be amplified via our positional SNP-to-Gene mapping [47], which is foundational for identifying causal genes from CVs [39] but does not capture the full complexity of gene regulation. To maximize the accuracy of positional SNP-to-Gene mapping, we chose only to examine CVs that fell directly within gene regions. However, excluding intergenic variants may have excluded a non-random subset of functional variants, which could have skewed the observed convergence results. Future studies could leverage refined methods for SNP-to-gene mapping, such as those leveraging linkage disequilibrium [48], to improve the accuracy of input genes."

Figure S1. Further exploration of gene properties, disease groups, and synthetic associations. **a** Comparison of CVG and RVG properties across 373 traits using alternative metrics of gene constraint and gene size. The value for each set of CVGs or RVGs was taken as the median of all genes in each set (Wilcoxon Signed-Rank Test, BH correction).

The center box plots show the median property value and interquartile range (IQR), with the lower and upper whiskers extending to $Q1 - 1.5IQR$ and $Q3 + 1.5IQR$. The violins extend to the minimum and maximum observations. Red lines indicate the median value for all protein-coding genes. *** $q < 1 \times 10^{-5}$.

- This is another stylistic comment, so the authors can feel free to ignore, but I found the nomenclature of "trait network" to be a bit confusing. It's not a network on traits (e.g., nodes are traits), and instead it's more about the overlap between genes that have network evidence from both CVGs and RVGs.

Thank you for bringing this confusion to our attention. We have modified the phrase "trait network" to "trait-specific network" throughout the manuscript to make it clearer that we are referring to a network generated for a trait, rather than a network on traits.

- Figure 4A suggests that maybe overlap is being driven (at least in part) by power. Does a similar trend hold when using the number of genes discovered by GWAS instead of heritability? This would be consistent with <https://doi.org/10.1101/2024.12.12.628073>, where it's suggested that GWAS discovers an essentially random subset of the set of genes that contribute to a trait, and so increasing the number of genes discovered by GWAS would increase the likelihood/degree of overlap.

We have added **new Supplemental Figure S3**, which we reference in the results section *Heritability and gene properties drive network convergence*:

"The COLOC scores for these subsets of traits were strongly correlated with trait heritability ($r_s = 0.61$, **Figure 4a**), but not population prevalence ($r_s = -0.01$, **Figure 4b**). We further observed that classes of traits with the highest heritability, such as lipid measurement and metabolic traits, also showed the strongest convergence of common and rare variant associations ($r_s = 0.83$, **Figure S3a**). To a lesser extent, the COLOC score was correlated with the number of associated genes, both within biological domains ($r_{CVG} = 0.31$, $r_{RVG} = 0.35$, **Figure S3a**) and globally ($r_{CVG} = 0.32$, $r_{RVG} = 0.28$, **Figure S3b**). These results suggest that both higher heritability and study power may increase the observed network convergence between common and rare variant associations."

Figure S3. Network convergence as a function of gene set and biological properties. **a** Correlation of mean COLOC score across biological domains with trait SNP heritability, number of CVGs, and number of RVGs. Points represent the mean values within each domain, with error bars representing the standard error. Point size indicates the number of traits per domain. Heritability results are shown for the subset of 197 traits for which a SNP heritability estimate was available, and the number of gene results are shown for the 373 best trait study pairs (Figure S2b). **b** COLOC score as a function of the number of underlying CVG and RVG associations. Total association count is the total number of distinct CVGs and RVGs per trait. Spearman correlation reported. Line plots show the mean COLOC score for each quantile, and violin plots show the distribution of COLOC scores within each feature quantile, with the median and quartiles represented by horizontal lines. Violins extend to the minimum and maximum observed values.

- It would be good to include a caveat about the significance measure in Table 1. In particular, I suspect that many of these predictors are highly correlated (and hence nearly collinear), and so the p-values / confidence intervals reflect not only the uncertainty in the marginal association, but the extent to which any association can be explained away by other predictors. E.g., "Number of genes" might have a large effect, but if it correlates with "Categorical trait", then many elastic net models will drop one predictor or the other.

We have added an updated reference to the existing figure **Supplemental Figure S4a** (formerly Supplemental Figure 2a) which shows the correlations among the test features to the results section *Heritability and gene properties drive network convergence*:

"However, due to widespread correlations between the features (**Figure S4a**), the reported coefficients and confidence intervals may partly reflect shared variance among related features. Therefore, the importance of properties such as the number of associations and the study population size cannot be ruled out."

- It would be good to include a discussion of <https://doi.org/10.1101/2025.02.24.639925> which also investigates the convergence of rare and common variant.

See next comment.

Reviewer #1 (Significance (Required)):

This is an important paper and provides new insights onto a problem that has been of recent interest to the statistical genetics community. The network approach to the problem is novel and a strength of the paper. The paper could be improved by better connecting its results to similar papers in the field (e.g., <https://doi.org/10.1186/s13073-023-01253-9>, <https://doi.org/10.1038/s41586-022-05684-z>, <https://doi.org/10.1101/2024.12.12.628073>, and <https://doi.org/10.1101/2025.02.24.639925>), and there are a few (easily fixable) potential technical flaws with the work. The paper should be interesting to a broad segment of the statistical genetics / human genetics community, and should be useful in the interpretation and design of association studies.

For context, my expertise is in statistical genetics, population genetics, polygenic scores, and statistics.

We have expanded our *Discussion* to include comparisons to other related works, including <https://doi.org/10.1186/s13073-023-01253-9>, <https://doi.org/10.1038/s41586-022-05684-z>, <https://doi.org/10.1101/2024.12.12.628073>, and <https://doi.org/10.1101/2025.02.24.639925>)

“Each genetic study design likely detects a subsample of the complete set of causal genes, influenced by varying biological and technical factors. Consistent with this premise, we observe significant differences between CV-associated genes (CVGs) and RV-associated genes (RVGs) in terms of structure, expression, and conservation (**Figure 1c**). These differences likely drive the low concordance of implicated genes. While some studies report the coincidence of RVs at GWAS loci as high as 70-97% [3,6,36–38], others have reported that only 26% of RVs are contained within top GWAS loci [7] with low concordance in the implicated genes [3,7,39]. Recently, Zhou et al. defined the COmmon variant and RAre variant Convergence (CORAC) signature to assess the gene-level overlap of CVGs and RVGs, finding higher CORAC estimates for continuous traits with large effective sample sizes. In comparison, traits with high polygenicity and stronger negative selection produced lower CORAC estimates [8]. Taken together, these findings suggest that the observed overlap of CVGs and RVGs is limited by incomplete study power, the complexity of genetic architectures, and the influences of selection on observed variants.

Under these conditions, looking beyond individual loci and genes is essential for understanding the joint role of CVs and RVs in complex traits. Wiener et al. showed that CVs and RVs implicate similar cell types and have pleiotropic effects on similar traits, suggesting mechanistic convergence of variants at a systems level [40]. We build on these results by integrating CVs and RVs within a comprehensive biological network to enable both quantification and interpretation of the systems-level convergence (**Figure 2a**). Across a range of biological contexts, networks have aided in identifying convergent complexes and pathways from disparate genetic data, including for the study of CVs and RVs in select traits [11,12]. Our study substantially expands the scope of these applications to assess network convergence of variants across a diverse range of human traits, including association studies performed in various human

populations. In doing so, we confirm the convergence of disparate variants to consistent underlying processes for the majority of traits (**Figure 2**) and identify factors driving variability in this convergence (**Figure 4**). “

And:

“The finding that traits with high SNP heritability exhibit stronger network convergence (**Figure 4a**) builds on previous findings that common and rare variant heritability are correlated [40] and that incorporating rare variants enhances heritability prediction [5]. Using the CORAC signature, greater convergence of highly powered quantitative traits was similarly observed [8], but highly polygenic traits had dampened convergence. In contrast, we found that traits with more numerous association signals tended to show stronger network convergence (**Figure S3**). While factors such as study power and design confound the number of genes identified, this result highlights the advantage of a network approach for capturing convergence in complex genetic architectures.”

Reviewer #2 (Evidence, reproducibility and clarity (Required)):

The manuscript of Wright et al describes a very important phenomenon, how common and rare variants are connected in molecular networks across multiple disease traits. The manuscript text is clear with a good flow. The figures present complex analyses in a visually understandable way with relevant and helpful captions. The authors executed a state-of-the-art network biology analysis with a well-documented, community standard and novel workflow. The output data (networks) was also shared as expected based on best practices. In each analytical step, adequate statistical analyses describe key features and properties of their finding. The Discussion is surprisingly short compared to the complexity and length of the Result section. Limitations of the study are embedded in generally positive short paragraphs. Comparison to similar approaches are missing.

We have significantly expanded our *Discussion* to include comparisons to other important works:

“Each genetic study design likely detects a subsample of the complete set of causal genes, influenced by varying biological and technical factors. Consistent with this premise, we observe significant differences between CV-associated genes (CVGs) and RV-associated genes (RVGs) in terms of structure, expression, and conservation (**Figure 1c**). These differences likely drive the low concordance of implicated genes. While some studies report the coincidence of RVs at GWAS loci as high as 70-97% [3,6,36–38], others have reported that only 26% of RVs are contained within top GWAS loci [7] with low concordance in the implicated genes [3,7,39]. Recently, Zhou et al. defined the COmmon variant and RAre variant Convergence (CORAC) signature to assess the gene-level overlap of CVGs and RVGs, finding higher CORAC estimates for continuous traits with large effective sample sizes. In comparison, traits with high polygenicity and stronger negative selection produced lower CORAC estimates [8]. Taken together, these findings suggest that the observed overlap of CVGs and RVGs is limited by incomplete study power, the complexity of genetic architectures, and the influences of selection on observed variants.

Under these conditions, looking beyond individual loci and genes is essential for understanding the joint role of CVs and RVs in complex traits. Wiener et al. showed that CVs and

RVs implicate similar cell types and have pleiotropic effects on similar traits, suggesting mechanistic convergence of variants at a systems level [40]. We build on these results by integrating CVs and RVs within a comprehensive biological network to enable both quantification and interpretation of the systems-level convergence (**Figure 2a**). Across a range of biological contexts, networks have aided in identifying convergent complexes and pathways from disparate genetic data, including for the study of CVs and RVs in select traits [11,12]. Our study substantially expands the scope of these applications to assess network convergence of variants across a diverse range of human traits, including association studies performed in various human populations. In doing so, we confirm the convergence of disparate variants to consistent underlying processes for the majority of traits (**Figure 2**) and identify factors driving variability in this convergence (**Figure 4**). “

And:

“The finding that traits with high SNP heritability exhibit stronger network convergence (**Figure 4a**) builds on previous findings that common and rare variant heritability are correlated [40] and that incorporating rare variants enhances heritability prediction [5]. Using the CORAC signature, greater convergence of highly powered quantitative traits was similarly observed [8], but highly polygenic traits had dampened convergence. In contrast, we found that traits with more numerous association signals tended to show stronger network convergence (**Figure S3**). While factors such as study power and design confound the number of genes identified, this result highlights the advantage of a network approach for capturing convergence in complex genetic architectures.”

And more clearly state the limitations of the study:

“Several limitations should be considered when interpreting these results. Some distinct properties of CVGs and RVGs may result from underlying selection biases and limitations of the SNP-to-Gene mapping procedure. For example, larger genes have more opportunities to be tagged by a causal variant in GWAS, which is reflected in our finding that CVGs tended to be longer (**Figure 1c**). This bias may be amplified via our positional SNP-to-Gene mapping [47], which is foundational for identifying causal genes from CVs [39] but does not capture the full complexity of gene regulation. To maximize the accuracy of positional SNP-to-Gene mapping, we chose only to examine CVs that fell directly within gene regions. However, excluding intergenic variants may have excluded a non-random subset of functional variants, which could have skewed the observed convergence results. Future studies could leverage refined methods for SNP-to-gene mapping, such as those leveraging linkage disequilibrium [48], to improve the accuracy of input genes. Lastly, while our results demonstrate the mechanistic convergence of common and rare variants, this does not imply their co-occurrence within individuals. Understanding the correlation or mutual exclusivity of convergent variants and systems across populations could inform the stratification of individuals based on overall genetic risk and underlying genetic architecture.”

Here, I would like to raise a few points to increase the validity of the findings and improve the biological insight we can gain from this work:

- The network analysis work is based on the main author's previous integrated and combined network resource, PCNet2. While this is an excellent resource for protein-protein interactions, some key analysis should be also done as a control with other network sources (for example

using only one key component of PCNet2 (eg STRING) and one similar large network not part of PCNet 2 (eg OmniPath).

We have now performed the main network colocalization analysis using OmniPath alongside our existing analysis with the PCNet 2.0 subcomponents PCNet 2.2, HumanNet, and STRING High Confidence networks. In addition to the correlation between COLOC score results, we also now present the total fraction of traits identified as convergent using each network. These results are presented in **expanded Supplemental Figure S2c** and **new Supplemental Figure S2d**. We have expanded our reference to these findings in the results section *Network convergence is robust across variable study designs* as:

“The observed convergence of CVGs and RVGs remained consistent when the network colocalization analysis was performed using subcomponents of PCNet 2.0 (**Figure S2c**). The networks HumanNet v3 [24], STRING [25], and the co-citation-free PCNet 2.2 [17] generated COLOC scores with Spearman correlations of 0.93, 0.92, and 0.95 to PCNet 2.0, respectively. Using an independent large network, OmniPath [26], we observed a slightly reduced performance with 68% of traits showing significant network convergence, compared to an average of 75% for PCNet 2.0 and its subcomponents (**Figure S2d**). Overall, while the global trend of CVG/RVG convergence is robust to network selection, the results for individual traits may vary based on the underlyingly network.”

And we have added details for sourcing OmniPath in the Methods section *Data Curation – Biological Knowledge Networks*:

“We defined the OmniPath network using the omnipath Python client [17], sourcing all interactions except those involving small molecules. We binarized all multiplex interactions and standardized the networks using the Python package neteval v0.2.2 [46] to remove duplicates and convert identifiers to NCBI Gene IDs. All interactions were treated as undirected.”

Figure S2. Robustness of network convergence across different studies and networks. **c** Pairwise comparison of COLOC scores across five biological networks. Correlation (r) calculated using Spearman correlation. The number of traits that could be assessed by both networks is given by n . **d** For each network, the fraction of evaluated traits with a nominally significant COLOC score ($p < 0.05$). Related to Figure 3.

- It should be clarified throughout that this work focuses on coding region variants (SNPs), and, as explained in the Discussion very briefly, left out non-coding SNPs - which are the majority

of common and rare variants. Because this understandable technical filter, the conclusions and main messages should be accordingly toned down.

We have clarified this point in the results section *Identification of 373 human traits with common and rare variant associations*:

“Common-variant-associated genes (CVGs) were defined as distinct genes with a trait-associated single-nucleotide polymorphism (SNP) located within the gene region defined by Ensembl [13,15].”

And in the results section *A majority of traits demonstrate network convergence of common and rare variant associations*:

“Of the 373 traits, 254 showed significant network convergence of CVGs and RVGs (**Figure 2b**), suggesting a high concordance in the molecular systems impacted by common and rare variants present in gene regions.”

And in the *Discussion*:

“To maximize the accuracy of positional SNP-to-Gene mapping, we chose only to examine CVs that fell directly within gene regions. However, excluding intergenic variants may have excluded a non-random subset of functional variants, which could have skewed the observed convergence results. Future studies could leverage refined methods for SNP-to-gene mapping, such as those leveraging linkage disequilibrium [48], to improve the accuracy of input genes.”

- It would be interesting to add some thoughts to the Discussion how this analysis could be done with gene regulatory networks, and whether the authors expect that the identified phenomena is limited to protein-protein interaction networks, or will probably true for regulatory networks as well.

We have clarified the evidence types included in the network used in the results section *A majority of traits demonstrate network convergence of common and rare variant associations*:

“...we integrated CVGs and RVGs with a comprehensive biological knowledge network, the Parsimonious Composite Network (PCNet 2.0) [17]. This network contains 3.85 million pairwise relationships among human genes and proteins derived from a variety of evidence sources such as protein-protein interactions, co-expression, and known biological pathways.”

And have added the following paragraph to the *Discussion*:

“Similarly, some traits showed variable results with different networks, possibly due to the varying representation of systems and datatypes within each resource [17]. Network convergence could also be evaluated using NetColoc with targeted network resources, such as regulatory or signaling-specific networks [17,45,46]. Implementing NetColoc with these resources will require additional adaptation and benchmarking to leverage the directed biological relationships present in some networks. Ultimately, we anticipate that such applications will elucidate mechanism-specific subsets of convergent genetic variation.”

- The authors analysed tissue specificity and Human Protein Atlas based tissue-specific

enrichment, but given the currently already available single-cell datasets from some of the published Human Cell Atlas projects, a more fine-grained analysis would be helpful. Such large cell-type specific expression data overlaid with the networks would be important to dissect and check whether the observed network colocalizations are not the product of aggregated (tissue-level) interactions.

We agree that using single-cell expression data could provide a more detailed analysis of gene and variant specificity within complex traits. However, we feel a dedicated study would be best for dissecting differences between cell-type and aggregated tissue-level interactions in the context of network colocalization. In addition, many of the complex traits included in our study are expected to manifest over multiple cell types and even tissues. We have noted the limitations of using tissue-level data in the *Discussion*:

“In this study, we utilized bulk-tissue expression to capture broad patterns of gene activity, meaning that the observed enrichments may reflect aggregate effects across multiple cell types rather than convergence within specific cell populations. Indeed, differences in the specificity of regulatory effects between variant classes have been reported: Li et al. reported that RV expression quantitative trait loci (eQTLs) are more tissue-specific than CV eQTLs [43], whereas Cuomo et al. found that CV eQTLs are more cell-type-specific than RV eQTLs [44]. Future integration of eQTLs and single-cell expression data alongside common and rare variants could clarify the roles of tissue-specific and cell-type-specific variants.”

- At multiple analyses, interesting differences between disease groups (eg infection, cancer, neurodegeneration) are pointed out. However, these differences are not explored or explained later. This is a missed opportunity to demonstrate the applicability of the authors' work to gain new biological insight on the systems-level features of various diseases. Some comparative results with explanation, and a new paragraph in the Discussion could cover this.

We have added **new Supplemental Figure S1b** and **new Supplemental Figure S3a** to assess the differences in trait features across the different disease groups. We present these findings in the results section *A majority of traits demonstrate network convergence of common and rare variant associations*:

“Network convergence was particularly strong among lipid measurement traits, which had trait-specific networks that were, on average, 3.6 times larger than expected by chance. In contrast, infection, neoplasm, and skeletal traits exhibited low rates of network convergence, with significant COLOC scores for only 4 of 18, 9 of 26, and 6 of 17 traits, respectively (**Figure 2d**). The biological domains with stronger convergence contained a higher proportion of continuous traits, and these traits tended to have a higher number of shared genes ($r_s = 0.82$, **Figure S1b**).”

And in *Heritability and gene properties drive network convergence*:

“We further observed that classes of traits with the highest heritability, such as lipid measurement and metabolic traits, also showed the strongest convergence of common and rare variant associations ($r_s = 0.83$, **Figure S3a**). To a lesser extent, the COLOC score was correlated with the number of associated genes, both within biological domains ($r_{CVG} = 0.31$, $r_{RVG} = 0.35$, **Figure S3a**) and globally ($r_{CVG} = 0.32$, $r_{RVG} = 0.28$, **Figure S3b**). These results suggest that both higher

heritability and study power may increase the observed network convergence between common and rare variant associations.”

Figure S1. Further exploration of gene properties, disease groups, and synthetic associations. **b** Comparison of the COLOC score and the number of shared CVG and RVG genes for traits within each biological domain. Points represent the mean values within each domain, with error bars representing the standard error. Point size indicates the number of traits per domain.

Figure S3. Network convergence as a function of gene set and biological properties. **a** Correlation of mean COLOC score across biological domains with trait SNP heritability, number of CVGs, and number of RVGs. Points represent the mean values within each domain, with error bars representing the standard error. Point size indicates the number of traits per domain. Heritability results are shown for the subset of 197 traits for which a SNP heritability estimate was available, and the number of gene results are shown for the 373 best trait study pairs (Figure S2b).

****Referees cross-commenting****

I agree with the points raised by the other reviewers. Also, good to see that all the three of us felt it is a good study that needs a bit of revision but not substantial.

Reviewer #2 (Significance (Required)):

Overall, this is a great manuscript and well conducted study. The core concept is very important for any researcher working on systems genomics, network biology and higher-level diseases systems. As the work follows best practices in systems biology, the paper will be very interesting for researchers at various career stages and for teaching/training purposes as a good example study.

We appreciate the reviewer's positive assessment of our work and feel that the comments have improved the clarity and impact of our manuscript.

Reviewer #3 (Evidence, reproducibility and clarity (Required)):

The genetic architecture of complex human traits and diseases remains an open challenge. GWAS have been extensively used over the past decade to discover associations between common variants (typically MAF >1%) and human phenotypes, while whole-exome sequencing (WES) studies have been employed to identify such associations for rare variants. Several pioneering studies have demonstrated, for selected diseases, that associated common and rare variants can converge on shared biological processes. This study by Wright et al. is the first to systematically investigate this intriguing question. To address it, the authors applied a cutting-edge systems biology approach based on network colocalization. Analyzing 373 traits, they found that 254 showed significant network convergence between genes implicated by GWAS ("common variant genes" - CVGs) and by WES ("rare variant genes" - RVGs). This result is especially remarkable given that, for approximately 50% of the traits (189/373), there was no direct gene overlap between the CVG and RVG sets. The study deals with an important and unexplored problem. Overall, it is very well written, and the results are presented very well. A few issues must be addressed before it can be accepted.

Major Comments

- Shared genes between the CVG and RVG sets:

One key issue should be addressed regarding the analysis relating to the genes shared between the CVG and RVG sets. It is known that in some cases, common SNPs identified in GWAS may tag nearby causal rare variants ('synthetic associations'). In such cases, the shared genes may, in fact, be influenced solely by the rare variants. This is particularly important given the strong correlation observed between the number of shared genes for the trait and its COLOC score for the CVG and RVG networks. The authors should rule out this possibility.

We acknowledge that some of the shared genes will represent synthetic associations. For example, the well-documented synthetic association of *NOD2* in Crohn's Disease is present in our data. However, systematically identifying and excluding these associations is challenging using only the GWAS summary results and RV gene-level results available to us. To assess the potential effect of synthetic associations on the COLOC Score, we analyzed Crohn's Disease and 10 randomly selected traits with at least 3 shared genes. For each, we ran NetColoc, treating all shared genes as RVGs only. We have added **new Supplemental Figure S1c** to the

results section *A majority of traits demonstrate network convergence of common and rare variant associations:*

“The identification of some shared genes could be driven by synthetic associations [20], whereby a CV tags nearby causal RVs, rather than by the existence of independent causal RVs and CVs. When treating all shared genes as synthetic associations for a subset of traits, we found that significant convergence was maintained for 82% of tested traits, albeit with decreased COLOC scores (**Figure S1c**). Therefore, given the low expected frequency of synthetic associations [21–23], these associations are unlikely to be a significant driver of the observed CVG/RVG network convergence.”

Figure S1. Further exploration of gene properties, disease groups, and synthetic associations. **c** Estimation of the impact of synthetic associations on COLOC scores. Ten random traits with at least 3 shared genes identified by CV and RV association studies, and Crohn’s disease (a trait with a known synthetic association), were analyzed by removing shared genes from the set of CVGs. The top bar plot shows the number of shared genes for each trait, and the middle bar plot shows the shared genes as a percentage of all CVGs. The bottom bar plot shows the COLOC scores for each trait, allowing shared genes or treating shared genes as RVGs only. Colored bars represent significant network colocalizations ($p < 10^{-3}$).

- Sensitivity to the network structure

- o Previous works have shown that similar tasks (e.g., active module identification) highly depends on the structure of the network. To show the contribution of the network structure, please provide an analysis of the impact of the selected network on the main results. For example, gradually drop a larger fraction of random edges and see how it affects the results. Alternatively, gradually apply higher number of edges swaps to introduce higher noise into the network. More ideas were applied in PMID: 33782690

We agree that it is important to show how our method utilizes the network information. We have now implemented a topology-preserving network randomization procedure to generate progressively noisier versions of two networks: PCNet 2.0 and STRING-High Confidence. We

perform NetColoc using these networks with the already defined partitioned GWAS testing traits, finding that the COLOC score decays as the network is randomized. We have added **new Supplemental Figure S8c-d**, which we describe in the *Methods* section *Optimization and Benchmarking of Q-NetColoc*:

“Input network dilution testing. To understand the importance of network structure, we tested the response of NetColoc to the introduction of random error in the networks. We performed degree-preserving network swaps using `shuffle_networks` from `neteval` [59] to create a series of progressively more randomized versions of PCNet 2.0 and STRING-HC with equivalent topologies. The extent of randomization (dilution) was measured based on $1-J$, where J is the Jaccard similarity of edges between the original and shuffled network. For PCNet 2.0, we generated three series of nine networks with dilution fractions between 0.18 to 0.8. For STRING-HC, we generated three series of nine networks with dilution fractions ranging from 0.18 to 0.95. We performed Q-NetColoc with $\beta = 20$, $\tau = 1$, and $\tau^* = 3$ for the partitioned GWAS test gene sets defined by S2G mapping, using a threshold of $\theta = 1 \times 10^{-5}$, and filtered to those with significant colocalization using the original network. We observed that COLOC scores consistently decayed as the network structures were randomized, indicating that the NetColoc procedure depends on the biological knowledge encoded in the edges, rather than simply on the network topology (**Figure S8c-d**).”

Figure S8. Benchmarking of Network Colocalization. **c-d** COLOC scores for partitioned GWAS test traits across randomized versions of (c) PCNet 2.0 and (d) STRING-HC. Test traits were filtered to those with a significant network colocalization using the original network (PCNet 2.0: $n = 37$, STRING-HC: $n = 28$). For each network, three series were generated by progressively diluting the original network (dilution = 0%) via degree-preserving edge swaps. Dilution percent is determined as $100 \times (1 - J)$, where J is the Jaccard similarity of edges in the original and randomized networks. Box plots show the median COLOC score and interquartile range (IQR), with the lower and upper whiskers extending to $Q1 - 1.5IQR$ and $Q3 + 1.5IQR$. Top bar plots show the number of test inputs displaying significant network colocalization (COLOC $P < 0.05$).

o In the linear regression estimates, the subnetwork density seems to be important in the prediction, and that density is also highly affected by the network structure. Systematic evaluation of different networks is relevant for this part as well.

We have also added **new Supplemental Figure S4d** to the results section *Heritability and gene properties drive network convergence*:

“We further examined the relationship between the interconnectedness and COLOC scores across progressively randomized network structures, where interconnectedness was defined as the density of CVG and RVG subnetworks. As interactions in PCNet 2.0 were randomized, the subnetwork densities and the correlations between the subnetwork densities and the COLOC scores were reduced (**Figure S4d**). Therefore, the network structure of the CVGs and RVGs plays an important role in determining the overall convergence.”

Figure S4. Expanded analysis of gene and trait features. **d** Subnetwork density of RVGs and CVGs, and Spearman correlation with COLOC score for partitioned GWAS test traits across progressively randomized versions of PCNet 2.0. Three series of randomized networks were generated by progressively diluting PCNet 2.0 via degree-preserving edge swaps, with subnetwork density calculated with each network structure. Dilution percent is determined as $100 \times (1 - J)$, where J is the Jaccard similarity of edges in the original and randomized networks. Bar plots show the median value across all three series, with error bars indicating the minimum and maximum values. Related to Figure 4.

- P-value justification.

The significance of the COLOC scores that underlie the study are based on computing COLOC score for 1000 randomized networks and then converting the Z-score a p-value using a assuming the distribution is standard normal. Please provide evidence that this assumption holds.

To show that this assumption holds, we have added new **Supplemental Figure S8e**, which we reference in the *Methods* section *Network Colocalization of Matched CVGs and RVGs using Q-NetColoc*:

“We examined the null distribution of expected trait-specific network size (used to determine the colocalization p-value) for a subset of test pairs with varying COLOC scores using quantile-quantile plots. This analysis confirmed that the null distributions of trait-specific network size approximated standard normal distributions (**Figure S8e**).”

Figure S8. Benchmarking of Network Colocalization. **e** Quantile-quantile plots for the expected distribution of colocalized network size for ten traits with varying COLOC score. All traits were ranked by COLOC score, with every 40th trait selected. For each trait, the expected distribution of sizes was calculated via 1000 permutations of the NPS scores and then standardized.

- Figure 2D: The dots are colored by significance (q-value). If so, how can some highly significant traits (dark purple) exhibit a COLOC score close to 1? Please clarify.

The colors in Figure 2d are intended to provide contrast between each domain and do not vary with colocalization significance. However, we acknowledge that the use of similar colors to Figure 2b invites this comparison. Therefore, we have **re-colored Figure 2d** as:

Figure 2. Network convergence of common and rare variant associations across 373 human traits. **d** Distribution of COLOC scores across biological domains. Purple lines show the median COLOC score per domain. The upper bar plot shows the percentage of domain traits that are continuous.

Other comments

- Gene prioritization:

"Compared to the original input genes, these prioritized trait genes and the full trait networks showed better enrichment for genes expressed in trait-relevant tissues, demonstrating that the network approach prioritizes functionally relevant genes (Figure 3F)": The term "prioritized" is confusing here since it was not defined. Please move the definition of the term from the legend to the main text where the term appears in the first time.

We agree and have modified the results section *Network convergence is robust across study designs* to define ‘prioritized’:

“Among the disjoint CVGs and RVGs genes, we defined prioritized genes as those included in the best-scoring trait-specific network for each trait. Overall, 44% of disjoint CVGs and 42% of disjoint RVGs were prioritized based on their global proximity to other trait-associated genes (**Figure 3e**).”

• Figure 5D:

o The enrichment analysis for the glucose measurement trait network uses the entire set of human genes as the background. It would be valuable to show whether the identified enriched processes are specific to the delineated network (e.g., whether trait-associated but non-prioritized genes show weaker enrichments).

We have added **new Supplemental Figure S5** to the results section *Tissue-specific RVGs are associated with stronger convergence*:

“Overall, the ten glucose measurement-associated genes included in the glucose measurement network showed enrichment for relevant pathways such as glucagon signaling ($p_{\text{adj}} = 3.4 \times 10^{-5}$) and galactose metabolism ($p_{\text{adj}} = 4.7 \times 10^{-5}$), as well as human phenotypes such as increased waist to hip ratio ($p_{\text{adj}} = 1.0 \times 10^{-5}$) (**Figure S5**). In contrast, the six non-prioritized trait-associated genes identified no significant enrichments for pathways or human phenotypes.

Globally, the ubiquitously expressed CVGs and pancreas-specific RVGs converged to crucial processes underlying glucose regulation (**Figure 5d**), such as AMPK signaling ($p_{\text{adj}} = 3.5 \times 10^{-29}$), glucagon signaling ($p_{\text{adj}} = 2.2 \times 10^{-28}$), insulin signaling ($p_{\text{adj}} = 1.5 \times 10^{-25}$), and insulin resistance ($p_{\text{adj}} = 7.5 \times 10^{-24}$). The glucose measurement network was also enriched for relevant human phenotypes such as an increased waist-to-hip ratio ($p_{\text{adj}} = 7.5 \times 10^{-20}$) and Type II Diabetes Mellitus ($p_{\text{adj}} = 1.4 \times 10^{-8}$). Therefore, integrating the glucose measurement associations within a biological network, rather than focusing solely on the CVGs and RVGs, prioritizes the most relevant genes and more fully captures the underlying biological processes.”

Figure S5. Pathway and phenotype enrichment of prioritized glucose measurement-associated genes. The top 10 KEGG pathway enrichments and all significant Human Phenotype Ontology (HP) enrichments are shown for the ten glucose measurement trait-associated genes that were prioritized by the network analysis. Of the ten genes, six were present in the KEGG and HP databases.

o What was the size of the glucose measurement trait network? From the fact that a threshold of >20 gene in the intersection was used it must be very large.

We have added this information to the caption of **Figure 5d**:

“**Figure 5.** Impacts of gene tissue distribution on network convergence of common and rare variant associations. **d** KEGG Pathway (top) and Human Phenotype Ontology (bottom) enrichments for the glucose measurement trait-specific network (n = 670 genes). Enrichments were filtered to those with adjusted p-value < 5×10^{-5} , term size < 1000, and gene intersection size > 20. Enrichments clustered based on Jaccard similarity of annotated genes using the Nearest Point Algorithm.”

• Figure 2c: Only a few categorical traits exhibit very high COLOC scores. What are the top 10 categorical traits? Do they share any common features that might explain their high scores?

The top categorical traits are not obviously delineated by features such as biological domain, number of CVGs/RVGs, or number of shared CVGs/RVGs. The high scores exhibited by these traits are likely the result of factors such as study power. We now identify the top categorical traits in **updated Figure 2c** in the results section *A majority of traits demonstrate network convergence of common and rare variant associations*:

“Continuous traits showed significantly stronger CVG/RVG convergence, consistent with the higher number of shared genes observed for these traits ($p = 1.0 \times 10^{-54}$, **Figure 2c**). Nonetheless, 74 categorical traits had significant convergence, with a diverse set of conditions represented in the top 10, including myeloproliferative disorder, Crohn’s disease, and allergic rhinitis.”

Figure 2. Network convergence of common and rare variant associations across 373 human traits. **c** Comparison of COLOC scores between categorical and continuous traits. The center box plots show the median property value and interquartile range (IQR), with the lower and upper whiskers extending to $Q1 - 1.5IQR$ and $Q3 + 1.5IQR$. The violins extend to the minimum and maximum observations. ILD: Interstitial Lung Disease, ASD: Autism Spectrum Disorder, DCM: Dilated Cardiomyopathy.

• Page 10: Two genes are reported with negative LOEUF scores. However, by definition, LOEUF scores are positive. Please clarify-were these scores processed or filtered incorrectly?

The reviewer is correct that the LOEUF values reported are incorrect. LoF Z-scores were mistakenly reported in place of LOEUF. We thank the reviewer for flagging this issue. Based on

the feedback received we have updated our analysis to use s_{het} (PMID: 38977852) as an alternative metric for gene constraint which is now included in the **updated Figure 1c**. We include analysis of the corrected LOEUF as well as evolutionary conservation (PhyloP) in the **new Supplemental Figure S1a**. The results section *Identification of 373 human traits with common and rare variant associations* now reads:

“Rare variant study populations tended to be larger, with a median of 210,000 individuals, compared to a median of 136,000 individuals for common variant studies (**Figure 1c**, $q = 6.8 \times 10^{-8}$). CVGs were more mutationally constrained ($q = 3.5 \times 10^{-29}$), as evidenced by higher values of s_{het} , a continuous metric that measures the reduction in fitness for heterozygous carriers of LOF mutations in a given gene [16]. In addition, CVGs had longer gene regions ($q = 1.6 \times 10^{-50}$) and more functional annotations ($q = 2.2 \times 10^{-5}$), with expression in a greater number of tissues ($q = 4.9 \times 10^{-4}$). The differences between CVGs and RVGs were consistent when examining alternative metrics of gene length and mutational constraint (**Figure S1a**). For example, CVGs had longer CDS sequences ($q = 8.4 \times 10^{-9}$) and showed stronger evolutionary conservation (phyloP, $q = 4.2 \times 10^{-11}$). These distinct gene properties likely reflect a combination of differences in variant impacts, as well as selection biases from the sequencing and statistical association methodologies.”

In addition, we have updated all our regression analysis and associated figures (**updated Figure 4, Supplemental Figure S4**) to use the s_{het} metric rather than LOEUF, and removed incorrect results from the results section *Tissue-specific RVGs are associated with stronger network convergence*. Details for the processing of the new metrics have been added to the Methods section *Data Curation*:

“Gene mutational constraint estimates based on loss-of-function (LoF) mutations were sourced from gnomAD v4.1 [54] (Constraint metrics TSV, <https://gnomad.broadinstitute.org/data>). For each gene, we extracted the LOEUF (loss-of-function observed/expected upper bound fraction) [54,55], the synonymous intolerance Z-score [55,56], and the expected number of LoF mutations. The LOEUF metric provides a continuous metric of a gene’s intolerance to loss-of-function variants, with a threshold of LOEUF < 0.6 recommended for defining LoF-constrained genes. For each distinct gene symbol, we prioritized entries associated with canonical and MANE Select transcripts, followed by prioritization of entries with an NCBI Gene ID assigned.

The lengths of coding sequences for each gene (CDS) were derived from GENCODE [57] v46 Basic Gene Annotation. Positional gene conservation scores (phyloP) were sourced from the UCSC Genome Browser [58], calculated via multiple alignment of 29 vertebrate species to the hg38 human genome. PhyloP scores from the reference chromosomes were aggregated for each transcript using BEDOPS v2.4.41 [59]. The final gene conservation score was defined per gene as the mean phyloP across all positions within the gene’s CDS.”

Figure 1. Curation and analysis of common and rare variant associations for 373 traits. **c** Comparison of CVG and RVG properties across 373 traits (Wilcoxon Signed-Rank Test, BH correction). For gene-level properties, the value for each set of CVGs or RVGs was taken as the median of all genes in each set. The center box plots show the median property value and interquartile range (IQR), with the lower and upper whiskers extending to $Q1 - 1.5IQR$ and $Q3 + 1.5IQR$. The violins extend to the minimum and maximum observations. Red lines indicate the median value for all protein-coding genes, where applicable. *** $q < 1 \times 10^{-5}$, ** $q < 1 \times 10^{-3}$, n.s. $q > 0.05$.

Figure S1. Further exploration of gene properties, disease groups, and synthetic associations. **a** Comparison of CVG and RVG properties across 373 traits using alternative metrics of gene constraint and gene size. The value for each set of CVGs or RVGs was taken as the median of all genes in each set (Wilcoxon Signed-Rank Test, BH correction). The center box plots show the median property value and interquartile range (IQR), with the lower and upper whiskers extending to $Q1 - 1.5IQR$ and $Q3 + 1.5IQR$. The violins extend to the minimum and maximum observations. Red lines indicate the median value for all protein-coding genes. *** $q < 1 \times 10^{-5}$.

- Table S2: There are entries where there is no overlap ($n_{\text{Shared}} = 0$) between CVGs and RVGs, yet the p-value for overlap (p_{Shared}) is significant. Please clarify.

We thank the reviewer for catching this inconsistency. We had an off-by-one error in our implementation of the hypergeometric test. We have now corrected and updated the affected results, including **Figure 1f** & **Supplemental Table 2**.

Figure 1. Curation and analysis of common and rare variant associations for 373 traits. **f** Stacked bar chart of the total number of distinct genes associated with each trait. Shared genes are those identified as both CVGs and RVGs. The top heatmap shows the q-value of the number of shared genes (Hypergeometric test, BH correction). Plot split into traits with > 100 associated genes and traits with ≤ 100 associated genes for visualization.

Reviewer #3 (Significance (Required)):

The study deals with an important and unexplored problem.

We found the analysis methodology novel and very interesting, more than the biological results.

We appreciate the reviewer's feedback and feel the comments given have improved and clarified our key findings.

Dear Prof. Ideker

First of all, Merry Christmas and happy holidays!

Thank you for the submission of your revised manuscript to our offices. We have now received the enclosed reports from the original referees at Review Commons that were asked to re-assess it for us at EMBO Reports. As you will see, EMBOR-2025-63223V1-T still has minor suggestions that I would like you to incorporate before we can proceed with the official acceptance of your manuscript.

Please note that things will be quite slow on our side during the holiday season, and most people will be back at the office only on early January, so there is no point in rushing your revision.

Yehu Moran
Academic Editor
EMBO Reports

Referee #1:

The authors addressed our main concerns. Below are a few remaining minor comments. These comments can be addressed by the authors without another review by us.

- Shared genes between the CVG and RVG sets

"When treating all shared genes as synthetic associations for a subset of traits, we found that significant convergence was maintained for 82% of tested traits"

The authors describe that significant convergence is retained for 82%. Such statement might be misleading for two reasons:

(1) From the text, it seems like all traits were tested, while, in fact, only 11 were (as described in the supp captions). The authors should simply write 9/11 tested traits

(2) The COLOC scores actually suffer substantial drop. Moreover, it seems like the drop is correlated with the % of overlap of the shared genes with CVG. This point should be discussed.

- Figure 5D

The number of genes here is extremely small (10 prioritized genes + 6 non-prioritized ones). Therefore, the differences in the enrichment analyses can be attributed to the difference in the number of genes in such small sets. Can the authors give other examples for traits with a larger number of prioritized and non-prioritized genes?

- Page 9+18:

"Our results indicated that the network convergence of CVGs and RVGs was stronger for traits with RVGs expressed in a limited number of tissues compared to CVGs (Figure 4H). To investigate the relevance of these genes, we performed a tissue-specific enrichment analysis for trait-relevant tissues, focusing on the traits exhibiting the largest differences in tissue distribution between the two sets of genes".

It is unclear how the four gene sets described in Methods under Enrichment for Tissue-Specific Genes were used to obtain a single enrichment score in the Tissue-specific enrichment analysis (TSEA). Please clarify.

Referee #2:

The authors have more than adequately addressed all of my previous comments, and as such I believe that the paper is suitable for publication and will make a nice addition to the literature.

Referee #3:

The authors comprehensively and adequately addressed my previous concerns. I don't have additional concerns. The network resource controls and the extended Discussion section now contain all the relevant, previously missing information.

Tamas Korcsmaros

Manuscript number: EMBOR-2025-63223V1-T

Corresponding author(s): Trey Ideker

Point-by-point description of revisions

Referee #1:

The authors addressed our main concerns. Below are a few remaining minor comments. These comments can be addressed by the authors without another review by us.

- Shared genes between the CVG and RVG sets "When treating all shared genes as synthetic associations for a subset of traits, we found that significant convergence was maintained for 82% of tested traits" The authors describe that significant convergence is retained for 82%. Such statement might be misleading for two reasons: (1) From the text, it seems like all traits were tested, while, in fact, only 11 were (as described in the supp captions). The authors should simply write 9/11 tested traits (2) The COLOC scores actually suffer a substantial drop. Moreover, it seems like the drop is correlated with the % of overlap of the shared genes with CVG. This point should be discussed.

We have added the detail requested (1) and clarified our interpretation of (2) in *A majority of traits demonstrate network convergence of common and rare variant associations*:

"The identification of some shared genes could be driven by synthetic associations (Dickson et al, 2010), whereby a CV tags nearby causal RVs, rather than by the existence of independent causal RVs and CVs. When treating all shared genes as synthetic associations for a subset of 11 traits, we found that significant convergence was maintained for 9 of the tested traits, but with decreased COLOC scores (**Fig. EV1C**). This decrease in colocalization strength is expected, as shared CVGs and RVGs (whether true independent signals or synthetic associations) strongly reinforce each other within the network analysis. Given the low expected frequency of synthetic associations (Anderson et al, 2011; Wray et al, 2011; Orozco et al, 2010), treating all shared genes as synthetic associations provides a conservative upper bound on their potential impact on CVG/RVG convergence."

Regarding (2), a correlation between the difference in COLOC score and the % of shared CVGs would not be surprising, as the % of shared CVGs can also be interpreted as a measure of strong reinforcing signals removed from the analysis. However, our analysis does not confirm a significant correlation ($r = -0.36$, $p = 0.28$; Spearman correlation).

- Figure 5D The number of genes here is extremely small (10 prioritized genes + 6 non-prioritized ones). Therefore, the differences in the enrichment analyses can be attributed to the difference in the number of genes in such small sets. Can the authors give other examples for traits with a larger number of prioritized and non-prioritized genes?

As previously requested, we included an analysis comparing enrichment results between prioritized and non-prioritized trait-associated genes for glucose measurement. We have

clarified our interpretation of this result in *Tissue-specific RVGs are associated with stronger network convergence* to address the set size caveat:

“Overall, the ten glucose measurement-associated genes included in the glucose measurement network showed enrichment for relevant pathways such as glucagon signaling ($p_{\text{adj}} = 3.4 \times 10^{-5}$) and galactose metabolism ($p_{\text{adj}} = 4.7 \times 10^{-5}$), as well as human phenotypes such as increased waist to hip ratio ($p_{\text{adj}} = 1.0 \times 10^{-5}$) (**Appendix Figure 1**). In contrast, the six non-prioritized trait-associated genes showed no significant enrichments for pathways or human phenotypes. However, it should be noted that the size of these gene sets may limit statistical power.”

We selected the glucose measurement trait for focused follow-up based on the strict criterion that it showed the greatest tissue expression difference between CVGs and RVGs. In the context of the analysis presented in this section, we do not believe that selecting additional traits based on post hoc criteria would be appropriate.

- Page 9+18:

"Our results indicated that the network convergence of CVGs and RVGs was stronger for traits with RVGs expressed in a limited number of tissues compared to CVGs (Figure 4H). To investigate the relevance of these genes, we performed a tissue-specific enrichment analysis for trait-relevant tissues, focusing on the traits exhibiting the largest differences in tissue distribution between the two sets of genes". It is unclear how the four gene sets described in Methods under Enrichment for Tissue-Specific Genes were used to obtain a single enrichment score in the Tissue-specific enrichment analysis (TSEA). Please clarify.

We have clarified the section **METHODS AND PROTOCOLS > Analysis and Interpretation of Common-Rare Trait-Specific Networks > Enrichment for Tissue-Specific Genes** to delineate the different TSEA analyses performed in this study, with references to the relevant figures. This section now reads:

“Tissue-specific enrichment analysis (TSEA) was performed using TissueEnrich (Jain & Tuteja, 2019), with all genes mapped to Ensembl gene identifiers. Tissue-specific genes were defined from RNA-seq results from the Human Protein Atlas (HPA (Uhlén *et al*, 2015)), included in TissueEnrich using default parameters of fold change ≥ 5 , minimum expression ≥ 1 , and number of expressed tissues ≤ 7 . For enrichment analysis, all genes classified as Tissue-Enriched, Tissue-Enhanced, or Group-Enriched were considered. The set of all genes in PCNet 2.0, converted to Ensembl gene identifiers, was used as the background gene list, and the BH correction was used to account for multiple hypothesis testing.

First, for all traits with significant network colocalization, we calculated the enrichment of the trait-specific networks for genes expressed in trait-relevant tissues (**Figure 3F**). For each trait, we defined four gene sets for each trait:

- All genes in the trait-specific network.

- Prioritized genes (input genes in the trait-specific network).
- All input genes (the union of CVGs and RVGs).
- Shared input genes (the intersection of CVGs and RVGs).

We then manually assigned relevant tissues based on the biological domain of each trait (e.g., Renal: Kidney). Where relevant tissues could not be clearly defined, traits were excluded from the analysis (e.g., anthropometric traits such as height and BMI). Enrichment analysis was performed using the R package TissEnrich v1.28.

Second, for the subset of traits with CVGs or RVGs expressed in a limited number of tissues, we examined the enrichment for genes expressed in trait-relevant tissues (**Figure 5B**). Traits were selected by the difference between the mean number of expressed tissues for CVGs and RVGs ($\delta_{nTissues}$), to give the 15 traits with the lowest $\delta_{nTissues}$ (RVGs expressed in fewer tissues) and the 15 traits with the highest $\delta_{nTissues}$ (CVGs expressed in fewer tissues). These traits were filtered to those that could be assigned a trait-relevant tissue based on the biological domain (as above), to give 23 traits. TSEA was performed for the input gene set (CVGs or RVGs) with expression in the fewest number of tissues on average. Enrichment analysis was performed using the R package TissEnrich v1.28.

Finally, TSEA and individual gene expression distributions for CVGs and RVGs associated with glucose metabolism were sourced from the TissueEnrich browser (<https://tissueenrich.gdcb.iastate.edu/>) using HPA gene expression data, utilizing gene symbols for the trait-associated genes and the background of PCNet 2.0 genes."

Referee #2:

The authors have more than adequately addressed all of my previous comments, and as such I believe that the paper is suitable for publication and will make a nice addition to the literature.

Referee #3:

The authors comprehensively and adequately addressed my previous concerns. I don't have additional concerns. The network resource controls and the extended Discussion section now contain all the relevant, previously missing information.

Tamas Korcsmaros

We thank all reviewers for their in-depth comments and suggestions, which have improved the quality of this manuscript.

Prof. Trey Ideker
University of California, San Diego
Department of Medicine
9500 Gilman Drive
La Jolla, CA 92093
United States

Dear Prof. Ideker,

I am very pleased to accept your manuscript for publication in the next available issue of EMBO reports. Thank you for your contribution to our journal.

You may qualify for financial assistance for your publication charges - either via a Springer Nature fully open access agreement or an EMBO initiative. Check your eligibility: <https://link.springer.com/journal/44319/how-to-publish-with-us>

Yours sincerely,

Yehu Moran
Academic Editor
EMBO Reports

>>> Please note that it is EMBO Reports policy for the transcript of the editorial process (containing referee reports and your response letter) to be published as an online supplement to each paper. If you do NOT want this, you will need to inform the Editorial Office via email immediately. More information is available here: <https://link.springer.com/partners/embo-press/editorial-policies#Peer%20review>